# A spiral attractor network drives rhythmic locomotion

**Angela M Bruno[1][†], William N Frost[2]\*, Mark D Humphries[3]\***

[1]Department of Neuroscience, The Chicago Medical School, Rosalind Franklin University of Medicine and Science, Illinois, United States; [2]Department of Cell Biology and Anatomy, The Chicago Medical School, Rosalind Franklin University of Medicine and Science, Illinois, United States; [3]Faculty of Biology, Medicine, and Health, University of Manchester, Manchester, United Kingdom

**Abstract** The joint activity of neural populations is high dimensional and complex. One strategy for reaching a tractable understanding of circuit function is to seek the simplest dynamical system that can account for the population activity. By imaging *Aplysia*'s pedal ganglion during fictive locomotion, here we show that its population-wide activity arises from a low-dimensional spiral attractor. Evoking locomotion moved the population into a low-dimensional, periodic, decaying orbit - a spiral - in which it behaved as a true attractor, converging to the same orbit when evoked, and returning to that orbit after transient perturbation. We found the same attractor in every preparation, and could predict motor output directly from its orbit, yet individual neurons' participation changed across consecutive locomotion bouts. From these results, we propose that only the low-dimensional dynamics for movement control, and not the high-dimensional population activity, are consistent within and between nervous systems.

**\*For correspondence:** william.frost@rosalindfranklin.edu (WNF); mark.humphries@manchester.ac.uk (MDH)

**Present address:** [†]Division of Biology and Biological Engineering, California Institute of Technology, Pasadena, California

**Competing interests:** The authors declare that no competing interests exist.

## Introduction

The increasing availability of large scale recordings of brain networks at single neuron resolution provides an unprecedented opportunity to discover underlying principles of motor control. However, such long-sought data sets are revealing a new challenge - the joint activity of large neural populations is both complex and high dimensional (*Ahrens et al., 2012*; *Cunningham and Yu, 2014*; *Yuste, 2015*). Population recordings have as many dimensions as neurons, and each neuron's activity can have a complex form. What strategies can we use to expose the hoped-for simplifying principles operating beneath the turbulent surface of real-world brain activity? One route is dimension reduction (*Briggman et al., 2006*; *Cunningham and Yu, 2014*; *Kobak et al., 2016*), which focuses on identifying the components of activity that co-vary across the members of a neural population, shifting the focus from the high dimensional recorded data to a low-dimensional representation of the population.

Such low-dimensional signals within joint population activity have been described in neural circuits for sensory encoding (*Mazor and Laurent, 2005*; *Bartho et al., 2009*), decision-making (*Briggman et al., 2005*; *Harvey et al., 2012*; *Mante et al., 2013*), navigation (*Seelig and Jayaraman, 2015*; *Peyrache et al., 2015*), and movement (*Levi et al., 2005*; *Ahrens et al., 2012*; *Kato et al., 2015*). Implicit in such dimension reduction approaches is the hypothesis that the high-dimensional population activity being recorded, while highly heterogenous, is derived from a simpler, consistent low-dimensional system (*Brody et al., 2003*; *Churchland et al., 2010*; *Kato et al., 2015*; *Miller, 2016*). We sought to directly test this hypothesis by identifying the simplest dynamical system that can account for high dimensional population activity.

**eLife digest** In all animals, neurons in the brain work together to generate movement. From a slug's ability to crawl, to your ability to move your hand, movement is dependent on hundreds or thousands of neurons being active at the same time. Rhythmic movements such as crawling or swimming show this clearly: groups of neurons fire together and remain silent together in a repeating sequence, producing waves of muscle contraction. But do we need to understand the activity of each of the hundreds or thousands of individual neurons to know how they generate these movements?

Bruno et al. argue that we do not, and propose instead that brain circuits that generate movement show a few set patterns of activity. By recording the activity of a population of neurons, we can identify the pattern of activity that generates a particular movement. To illustrate this point, Bruno et al. examined the network of neurons that drives the rhythmic crawling movement of the sea slug *Aplysia*.

The results show that the network of neurons seems to contain many different patterns of activity during crawling. Yet collectively these different patterns are reflections of a simpler hidden system, a spiral of ever-decreasing, oscillating activity. This pattern is referred to as a spiral attractor because whenever the network is activated, the overall pattern of activity is always pulled into this spiral regardless of its starting point. The same applies whenever the network is disturbed. The key thing to note, however, is that individual neurons within the network do not show the same activity each time the network is active. This means that only the spiral attractor itself, and not the contribution of the individual neurons, is constant every time the sea slug crawls.

What do we need to know to understand the brain? The results presented by Bruno et al. suggests that identifying the hidden systems that underlie seemingly complex and varying neural activity is the key to understanding how brains generate movement. This may also be true for how brains form memories, make decisions, and give rise to sight, hearing and touch.

A useful model to address these questions is the neural control of movement. Movement arises from the mass action of neuron populations (*Georgopoulos et al., 1986*; *Getting, 1989*; *Ahrens et al., 2012*; *Portugues et al., 2014*; *Yuste, 2015*; *Petersen and Berg, 2016*). While individual neuron activity can correlate with specific aspects of movement (*Chestek et al., 2007*; *Hatsopoulos et al., 2007*; *Churchland et al., 2010*, *2012*), the embedded low dimensional signals in population recordings (*Briggman et al., 2005*; *Levi et al., 2005*; *Kato et al., 2015*) and the intermittent participation of individual neurons across repeated movements in both vertebrates (*Carmena et al., 2005*; *Huber et al., 2012*) and invertebrates (*Hill et al., 2010*, *2015*) together suggest that only the collective population activity, and not specifics of single neuron firing, are key to movement control. If so, then finding the underlying dynamical system will be necessary for a parsimonious theory of the neural control of movement (*Briggman and Kristan, 2008*).

In order to identify the simplest dynamical system underlying population activity in movement control, we imaged large populations at single-neuron, single-spike resolution in the pedal ganglion of *Aplysia* during fictive locomotion (*Figure 1A*). The pedal ganglion presents an ideal target for testing hypotheses of movement control as it contains the pattern generator (*Jahan-Parwar and Fredman, 1979*, *1980*), motorneurons (*Hening et al., 1979*; *Fredman and Jahan-Parwar, 1980*) and modulatory neurons (*Hall and Lloyd, 1990*; *McPherson and Blankenship, 1992*) underlying locomotion. Moreover, its fictive locomotion is sustained for minutes, ideal for robustly characterising population dynamics. Using this model system, here we find its low-dimensional, underlying dynamical system, test if the low-dimensional signal encodes movement variables, and determine the contribution of single neurons to the low-dimensional dynamics.

We show that evoking fictive locomotion caused heterogenous population spiking activity, but under which always lay a low-dimensional, slowly decaying periodic orbit. This periodic trajectory met the convergence and perturbation criteria for an attractor. Crucially, we identify the attractor as a stable, decaying spiral in every preparation. We decoded motorneuron activity directly from the low-dimensional orbit, showing that it directly encodes the relevant variables for movement. Yet we

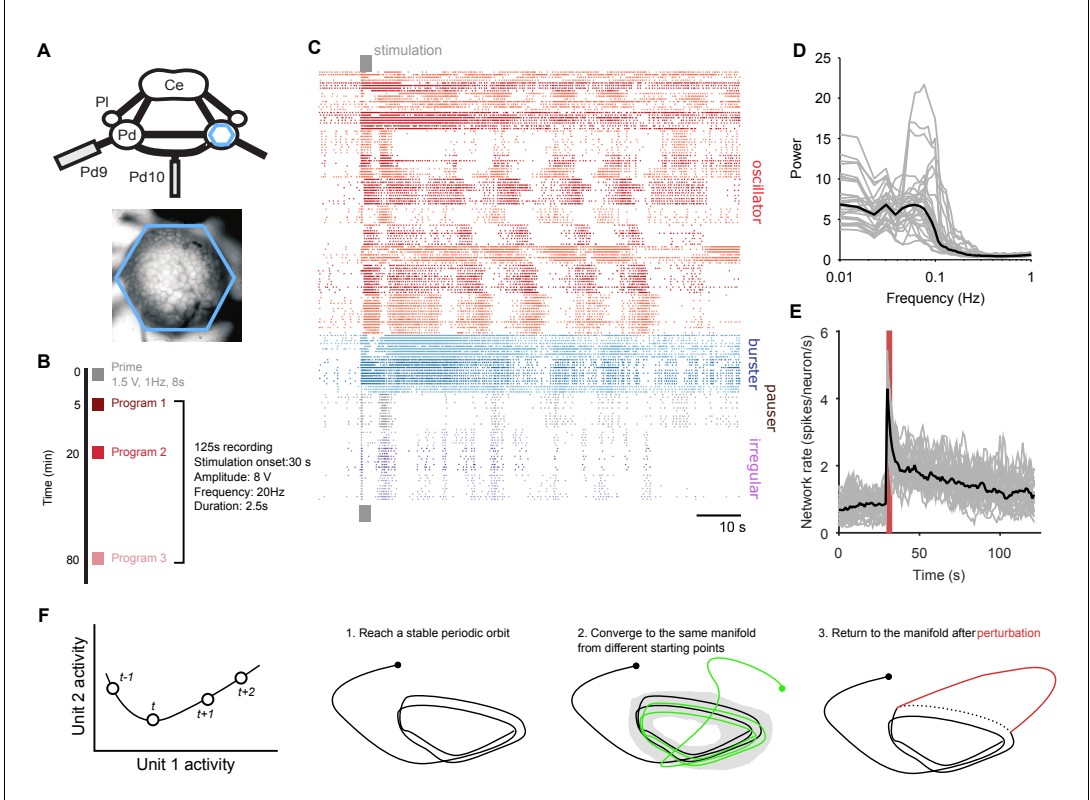

**Figure 1.** Population dynamics during fictive locomotion. (**A**) Voltage-sensitive dye recording of the pedal ganglion (Pd) network in an isolated central nervous system preparation (top) using a photodiode array (blue hexagon). The array covered the dorsal surface of the ganglion (bottom). Ce: cerebral ganglion; Pl: pleural ganglion; Pd9/10: pedal nerve 9/10. (**B**) Stimulus protocol. Three escape locomotion bouts were evoked in each preparation by stimulation of tail nerve Pd9. Parameters are given for the stimulus pulse train. (**C**) Example population recording. Raster plot of 160 neurons before and after Pd9 nerve stimulation. Neurons are grouped into ensembles of similarly-patterned firing, and ordered by ensemble type (colors) - see Materials and methods. (**D**) Power spectra of each population's spike-trains, post-stimulation (grey: mean spectrum of each bout; black: mean over all bouts). (**E**) Network firing rate over time (grey: every bout; black: mean; red bar: stimulation duration. Bins: 1 s). (**F**) Terminology and schematic illustration of the necessary conditions for identifying a periodic attractor (or 'cyclical' attractor). Left: to characterise the dynamics of a $N$-dimensional system, we use the joint activity of its $N$ units at each time-point $t$ – illustrated here for $N = 2$ units. The set of joint activity points in time order defines the system's trajectory (black line). Right: the three conditions for identifying a periodic attractor. In each panel, the line indicates the trajectory of the joint activity of all units in the dynamical system, starting from the solid dot. The manifold of a dynamical system is the space containing all possible trajectories of the unperturbed system – for periodic systems, we consider the manifold to contain all periodic parts of the trajectories (grey shading). In (condition 3), the dashed line indicates where the normal trajectory of the system would have been if not for the perturbation (red line). See *Figure 1—figure supplement 1* for a dynamical model illustrating these conditions.

The following figure supplement is available for figure 1:

**Figure supplement 1.** Necessary and sufficient conditions of a periodic attractor.

found that individual neurons varied their participation in the attractor between bouts of locomotion. Consequently, only the low-dimensional signal and not the high-dimensional population activity was consistent within and between nervous systems. These findings strongly constrain the possible implementations of the pattern generator for crawling in *Aplysia*; and by quantifying the attractor they make possible future testing of how short- and long-term learning change properties of that attractor. Collectively, these results provide experimental support for the long-standing idea that neural population activity is a high-dimensional emergent property of a simpler, low-dimensional dynamical system.

## Results

We sequentially evoked three bouts of fictive locomotion in each of 10 isolated central nervous system preparations (*Figure 1B*). Each bout of locomotion was evoked by short stimulation of the tail nerve P9, mimicking a sensory stimulus to the tail that elicits the escape locomotion response (*Hening et al., 1979*); in intact animals, a strong tail stimulus typically elicits a two-part escape behavior consisting of several cycles of a vigorous arching gallop, followed by several minutes of a more sedate rhythmic crawl (*Jahan-Parwar and Fredman, 1979*; *Flinn et al., 2001*). We imaged the dorsal pedal ganglion 30 s before through to 90 s after the evoking stimulus, aiming to capture the population dynamics initiating and driving the initial gallop before the transition to the crawl. Recorded populations from the pedal ganglion comprised 120–180 neurons each, representing ≈ 10% of the network in each recording. The population recordings captured rich, varied single neuron dynamics within the ganglion's network following the stimulus (*Figure 1C*). A dominant, slow (≤ 0.1 Hz) oscillation in neural firing (*Figure 1D*) is consistent with the periodic activity necessary to generate rhythmic locomotion. But the variety of single neuron dynamics (*Bruno et al., 2015*) (*Figure 1C*) and the slowly decaying population firing rate (*Figure 1F*) post-stimulus hint at a more complex underlying dynamical system driving locomotion than a simple, consistent oscillator.

Seeking the simplest dynamical system to account for these data, we first show here that the joint activity of the population meets the necessary conditions for a periodic attractor (*Figure 1F*). We identified these as: (1) applying a driving force causes the system's activity to fall onto a stable, periodic orbit; (2) repeatedly driving the system causes convergence of its activity to the same orbit; and (3) the system should return to the periodic orbit after the end of transient perturbation. *Figure 1— figure supplement 1* demonstrates these conditions in a dynamical model of a neural periodic attractor.

## Joint population activity forms a low-dimensional periodic orbit

We first established that under the heterogenous population activity evoked by the tail-nerve stimulation there was a low dimensional periodic trajectory, consistent with there being a periodic attractor in the pedal ganglion. Projections of a population's joint activity into three dimensions typically showed that stimulation caused a strong deviation from the spontaneous state, which then settled into repeated loops (*Figure 2A*). Capturing a significant proportion (80%) of the population variance generally required 4–8 embedding dimensions (*Figure 2B*), representing a dimension reduction by more than a factor of 10 compared to the number of neurons. Thus, throughout our analysis, we projected each evoked program into the number of embedding dimensions needed to capture at least 80% of the variance in population activity (4–8 dimensions: inset of *Figure 2B*). However, we cannot directly visualise this space; therefore we could not tell by visual inspection if the low-dimensional trajectory repeatedly returned to the same position, and so was truly periodic.

To determine whether population activity in higher dimensions reached a stable periodic orbit, we made use of the idea of recurrence (*Lathrop and Kostelich, 1989*; *Marwan et al., 2007*). For each time-point in the low-dimensional trajectory of the population's activity, we check if the trajectory passes close to the same point in the future (*Figure 2C*). If so, then the current time-point *recurs*, indicating that the joint activity of the population revisits the same state at least once. The time between the current time-point and when it recurs gives us the period of recurrence. A strongly periodic system would thus be revealed by its population's trajectory having many recurrent points with similar recurrence periods; random or chaotic dynamics, by contrast, would not show a single clustered recurrence period.

Plotting recurrent time-points showed that the evoked low-dimensional population activity typically recurred with a regular period (example in *Figure 2D*). We found strongly periodic recurrence on the scale of 10–15 s in many but not all of the 30 evoked population responses (*Figure 2E,F*). This reflected the range of stimulation responses from strongly periodic activity across the population to noisy, stuttering, irregular activity (*Figure 2—figure supplement 1*). Nonetheless, despite this heterogeneity across stimulus responses, the activity of almost all populations was dominated by a single periodic orbit (*Figure 2E*), robust to the choice of threshold for defining recurrence (*Figure 2—figure supplement 2*).

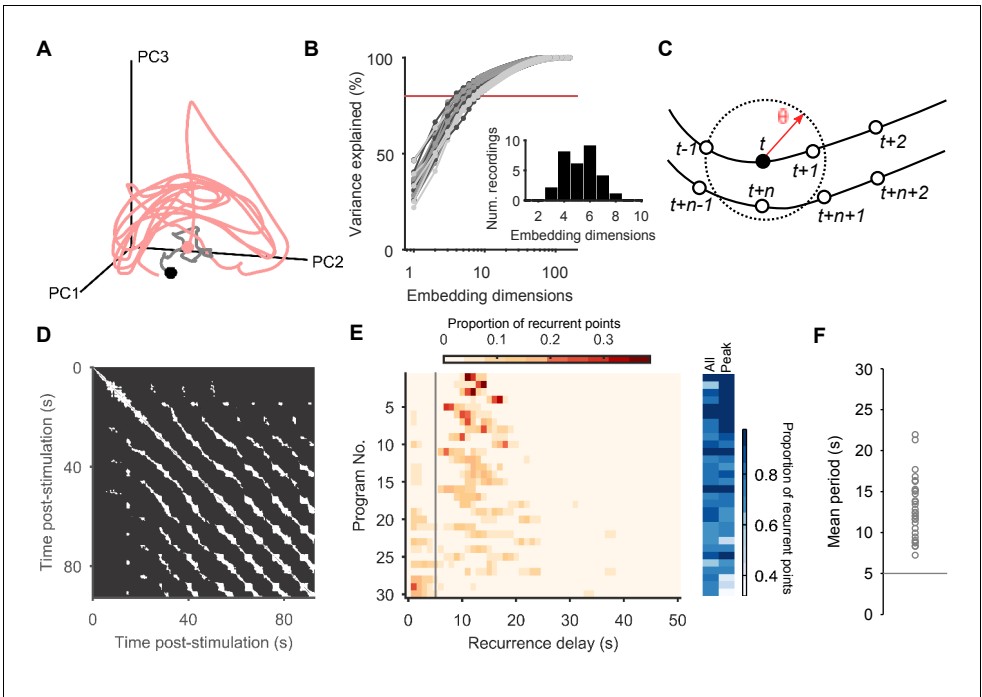

**Figure 2.** Population dynamics form a low-dimensional periodic orbit. (**A**) Projection of one evoked population response into three embedding dimensions, given by its first three principal components (PCs). Dots: start of recording (black) and stimulation (pink); spontaneous activity is shown in grey. Smoothed with 2 s boxcar window. (**B**) Proportion of population variance explained by each additional embedding dimension, for every evoked population response ($n = 30$; light-to-dark grey scale indicates stimulations 1 to 3 of a preparation). We chose a threshold of 80% variance (red line) to approximately capture the main dimensions: beyond this, small gains in explained variance required exponentially-increasing numbers of dimensions. Inset: Histogram of the number of PCs needed to explain 80% variance in every recorded population response. (**C**) Quantifying population dynamics using recurrence. Population activity at some time $t$ is a point in $N$-dimensional space (black circle), following some trajectory (line and open circles); that point *recurs* if activity at a later time $t + n$ passes within some small threshold distance $\theta$. The time $n$ is the recurrence time of point $t$. (**D**) Recurrence plot of the population response in panel A. White squares are recurrence times, where the low-dimensional dynamics at two different times passed within distance $\theta$. We defined $\theta$ as a percentile of all distances between points; here we use 10%. Stimulation caused the population's activity to recur with a regular period. Projection used 4 PCs. (**E**) Histograms of all recurrence times in each population response (threshold: 10%), ordered top-to-bottom by height of normalised peak value. Vertical line indicates the minimum time we used for defining the largest peak as the dominant period for that population response. Right: density of time-points that were recurrent, and density of recurrence points with times in the dominant period. (**F**) Periodic orbit of each evoked population response, estimated as the mean recurrence time from the dominant period.

The following figure supplements are available for figure 2:

**Figure supplement 1.** Range of dynamics in the evoked locomotion programs.

**Figure supplement 2.** Robustness of the periodic orbits.

## Joint population activity meets the conditions for a periodic attractor

The trajectory of a periodic dynamical system remains within a circumscribed region of space – the manifold – that is defined by all the possible states of that system. (We schematically illustrate a manifold by the grey shading in *Figure 1F* (condition 2), and demonstrate the manifold of our model periodic attractor network in panel C of *Figure 1—figure supplement 1*). If the population responses of the pedal ganglion are from an underlying periodic attractor, then the population's joint activity should rapidly reach and stay on its manifold when evoked; reach the same manifold

every time it is evoked; and return to the manifold when perturbed (these three conditions are schematically illustrated in *Figure 1F*; see *Figure 1—figure supplement 1* for the corresponding examples from the dynamical model).

We found that almost all evoked population responses quickly reached a state of high recurrence, within one oscillation period (*Figure 3A*), and were thereafter dominated by recurrence, indicating they quickly reached and stayed on the manifold.

But does each population response from the same preparation reach the same manifold? The key problem in analysing any putative attractor from experimental data is identifying when the

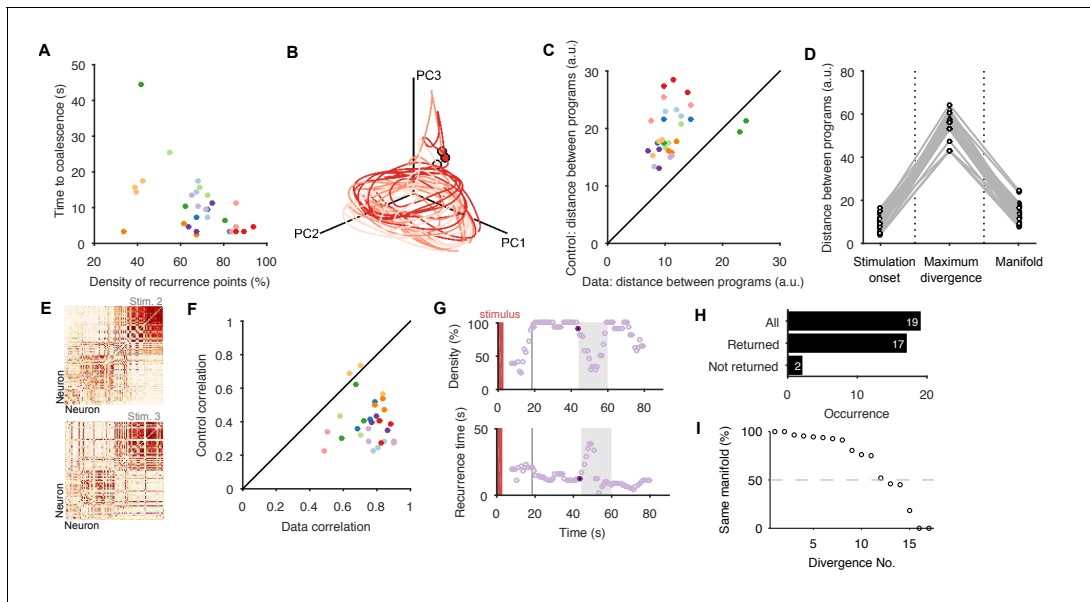

**Figure 3.** Low dimensional population dynamics meet the conditions for a periodic attractor. (**A**) Distribution of the time the population dynamics took to coalesce onto the attractor from the stimulation onset, and the subsequent stability of the attractor (measured by the proportion of recurrent points). Colours indicate evoked responses from the same preparation. The coalescence time is the mid-point of the first 5 s sliding window in which at least 90% of the points on the population trajectory recurred in the future. (**B**) Projection of three sequential population responses from one preparation onto the same embedding dimensions. Dots are time of stimulus offset. (**C**) Sequential population responses fall onto the same manifold. Dots indicate distances between pairs of population responses in the same preparation; color indicates preparation. Control distances are from random projections of each population response onto the same embedding dimensions - using the same time-series, but shuffling the assignment of time series to neurons. This shows how much of the manifold agreement is due to the choice of embedding dimensions alone. The two pairs below the diagonal are for response pairs (1,2) and (1,3) in preparation 4; this correctly identified the unique presence of apparent chaos in response 1 (see *Figure 3—figure supplement 1*). (**D**) Distances between pairs of population responses from the same preparation in three states: the end of spontaneous activity (at stimulus onset); between stimulation onset and coalescence (the maximum distance between the pair); and after both had coalesced (both reaching the putative attractor manifold; data from panel C). (**E**) Example neuron activity similarity matrices for consecutively evoked population responses. Neurons are ordered according to their total similarity in stimulation 2. (**F**) Correlation between pairs of neuron similarity matrices (Data) compared to the expected correlation between pairs of matrices with the same total similarity per neuron (Control). Values below the diagonal indicate conserved pairwise correlations between pairs of population responses within the same preparation. The two pairs on the diagonal are response pairs (1,3) and (2,3) in preparation 7; this correctly identified the unique presence of a random walk in response 3 (see *Figure 3—figure supplement 1*). (**G**) Spontaneous divergence from the trajectory. For one population response, here we plot the density of recurrence points (top) and the mean recurrence delay in 5 s sliding windows. Coalescence time: grey line. The sustained 'divergent' period of low recurrence (grey shading) shows the population spontaneously diverged from its ongoing trajectory, before returning. Black dot: pre-divergence window (panel I). (**H**) Breakdown of spontaneous perturbations across all population responses. Returned: population activity became stably recurrent after the perturbation. (**I**) Returning to the same manifold. For each of the 17 'Returned' perturbations in panel H, the proportion of the recurrent points in the pre-divergence window that recurred after the divergent period, indicating a return to the same manifold or to a different manifold.

The following figure supplements are available for figure 3:

**Figure supplement 1.** Convergence and non-convergence to the same manifold.

**Figure supplement 2.** Spontaneous perturbations of ongoing programs.

experimentally-measured dynamics are or are not on the attractor's manifold, whether due to perturbations of the system or noise in the measurements. Moreover, we cannot directly compare time-series between evoked responses because, as just demonstrated, each response may reach the manifold at different times (see also panel C in *Figure 1—figure supplement 1*). Thus the set of recurrent time-points allowed us to identify when the joint population activity was most likely on the attractor's manifold, and then to make comparisons between population responses.

To determine if sequentially-evoked responses from the same preparation reached the same manifold, we projected all 3 population responses into the same set of embedding dimensions, using only the recurrent points (*Figure 3B*; *Figure 3—figure supplement 1* shows these results are robust to other projections). Falling on the same manifold would mean that every recurrent point in one population response's trajectory would also appear in both the others' trajectories, if noiseless. Consequently, the maximum distance between any randomly chosen recurrent point in population response A and the closest recurrent point in population response B should be small. We defined small here as being shorter than the expected distance between a recurrent point in A and the closest point on a random projection of the activity in the same embedding dimensions. Despite the inherent noise and limited duration of the recordings, this is exactly what we found: pairs of evoked population responses from the same preparation fell close to each other throughout (*Figure 3C*), well in excess of the expected agreement between random projections of the data onto the same embedding dimensions.

We also checked that this convergence to the same manifold came from different initial conditions. The initiating stimulation is a rough kick to the system – indeed a fictive locomotion bout can be initiated with a variety of stimulation parameters (*Bruno et al., 2015*) – applied to ongoing spontaneous activity. Together, the stimulation and the state of spontaneous activity when it is applied should give different initial conditions from which the attractor manifold is reached. We found that the stimulation caused population responses within the same preparation to diverge far more than in either the spontaneous activity or after coalescing to the manifold (*Figure 3D*). Thus, a wide range of initial driven dynamics in the pedal ganglion population converged onto the same manifold.

Previous studies have used the consistency of pairwise correlations between neurons across conditions as indirect evidence for the convergence of population activity to an underlying attractor (*Yoon et al., 2013*; *Peyrache et al., 2015*). The intuition here is that neurons whose activity contributes to the same portion of the manifold will have simultaneous spiking, and so their activity will correlate across repeated visits of the population's activity to the same part of the manifold. To check this, we computed the pairwise similarity between all neurons within an evoked population response (*Figure 3E*), then correlated these similarity matrices between responses from the same preparation. We found that pair-wise similarity is indeed well-preserved across population responses in the same preparation (*Figure 3F*). This also shows that the apparent convergence to the same manifold is not an artefact of our choice of low-dimensional projection.

In many population responses, we noticed spontaneous perturbations of the low-dimensional dynamics away from the trajectory (examples in *Figure 3—figure supplement 2*), indicated by sudden falls in the density of recurrent points (*Figure 3G*). That is, perturbations could be detected by runs of contiguous points on the population trajectory that were not recurrent. As each spontaneous perturbation was a cessation of recurrence in a trajectory accounting for 80% of the co-variation between neurons, each was a population-wide alteration of neuron activity (see example rasters in *Figure 3—figure supplement 2*). In most cases (90%), the population dynamics returned to a recurrent state after the spontaneous perturbation (*Figure 3H*; *Figure 3—figure supplement 2*, panel B), consistent with the pertubation being caused by a transient effect on the population The two perturbations that did not return to a recurrent state were consistent with the end of the evoked fictive locomotion and a return to spontaneous activity (*Figure 3—figure supplement 2*, panel A). Of those that returned, all but three clearly returned to the same manifold (*Figure 3I*); for those three, the spontaneous perturbation appeared sufficient to move the population dynamics into a different periodic attractor (*Figure 3—figure supplement 2*, panel C). Potentially, these are the known transitions from the escape gallop to normal crawling (*Flinn et al., 2001*). The low dimensional dynamics of the pedal ganglion thus meet the stability, manifold convergence, and perturbation criteria of a periodic attractor network.

## Heterogenous population activity arises from a common attractor

While these results show the existence of a periodic orbit on an attractor in the evoked population responses, they cannot address whether these arise from the same putative attractor within and, crucially, between animals. To determine if there is a common underlying attractor despite the heterogeneity in spiking patterns across the population responses (*Figure 2—figure supplement 1*), we introduced a statistical approach to quantifying the low-dimensional trajectory. We first fitted a linear model of the local dynamics around each time point in the low-dimensional projection (see Materials and methods). For each $N$-dimensional point $P(t)$ in this projection, we fitted the $N$-dimensional model $\dot{P}* = \mathbf{A}P*$ to the trajectory forwards and backwards in time from point $P(t)$. In this model, the change in the trajectory over time $\dot{P}*$ in the neighbourhood of point $P(t)$ is determined by the values of the $N \times N$ matrix $\mathbf{A}$. The maximum eigenvalue of $A$ thus tells us whether the trajectory around point $P(t)$ is predominantly expanding or contracting in the $N$-dimensional projection, and whether or not it is rotating (*Strogatz, 1994*).

By fitting the linear model to each point on the trajectory we obtained time-series of the maximum eigenvalues, describing the local dynamics at each point along the trajectory. The time-series of eigenvalues typically showed long periods of similar magnitude eigenvalues, corresponding to the recurrent points (*Figure 4A*). Consequently, by then averaging over the eigenvalues obtained only for recurrent points, we could potentially capture the dynamics of the underlying attractor. Doing so, we found that the evoked population responses had highly clustered maximum eigenvalues (*Figure 4B,C*), and thus highly similar underlying dynamics despite the apparent heterogeneity of spike-train patterns between them. The dominance of negative complex eigenvalues implies the pedal ganglion network implements a contracting periodic orbit - it is a stable spiral attractor (*Figure 4D*).

In most population responses, the low-dimensional trajectory had negative, complex eigenvalues in all embedding dimensions, meaning that the spiral attractor completely characterised the population dynamics (*Figure 4—figure supplement 1*). Intriguingly, a few population responses had a positive real eigenvalue in one low-variance dimension (*Figure 4—figure supplement 1*), implying a simultaneous minor expansion of the population trajectory. This corresponded to the appearance of a small sub-set of neurons with increasing firing rates (*Figure 4E*).

The identification of a stable spiral makes a clear prediction for what should and should not change over time in the dynamics of the population. The negative complex eigenvalues mean that the magnitude of the orbit decays over time, corresponding to the decreasing population spike rate in most evoked responses (*Figure 1E*). However, a stable spiral indicates only a decrease in magnitude; it does not mean the orbital period is also slowing. Consequently, the presence of a stable spiral attractor predicts that the magnitude and period of the orbit are dissociable properties in the pedal ganglion network.

We checked this prediction using the linear model. The linear model estimated a mean orbital period of around 10 s (*Figure 4C*), consistent with the directly-derived estimate from the recurrent points (*Figure 2F*). This indicated the linear model was correctly capturing the local dynamics of each program. But our linear model also gave us a time-series of estimates of the local orbital period (*Figure 5A*), which we could use to check whether the orbital period was changing during each evoked response. We found that the population responses included all possible changes in periodic orbit: slowing, speeding up, and not changing (*Figure 5B*). As predicted there was no relationship between the contraction of the periodic orbit and its change in period (*Figure 5C*).

## The locomotion motor program can be decoded from the low-dimensional orbit

Collectively, these periodic, decaying dynamics are ethologically consistent with locomotion that comprises a repeated sequence of movements that decays in intensity over time (*Jahan-Parwar and Fredman, 1979*; *Flinn et al., 2001*; *Marinesco et al., 2004*). If this putative low-dimensional periodic attractor is the 'motor program' for locomotion, then we should be able to decode the locomotion muscle commands from its trajectory. In 3 of the 10 preparations we were able to simultaneously record activity from the P10 nerve that projects to the neck muscles (*Xin et al., 1996*) for all three evoked population responses. The spiking of axons in this nerve should correspond to the specific

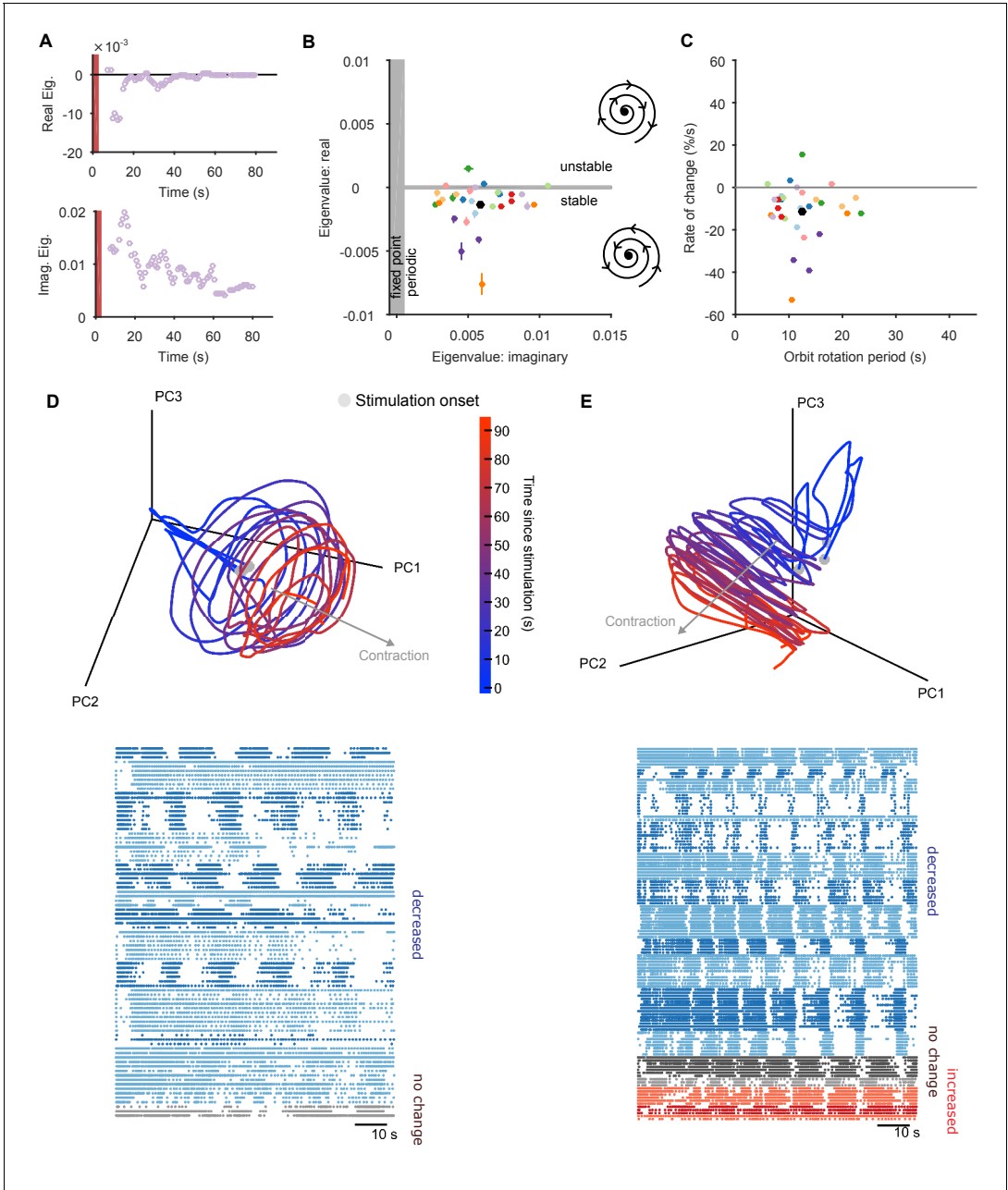

**Figure 4.** The pedal ganglion contains a spiral attractor. (A) Example time-series from one population response of the real (top) and imaginary (bottom) component of the maximum eigenvalue for the local linear model. Points are averages over a 5 s sliding window. Red bar indicates stimulus duration. (B) Dominant dynamics for each evoked population response. Dots and lines give means ±2 s.e.m. of the real and imaginary components of the maximum eigenvalues for the local linear model. Colours indicate responses from the same preparation. Black dot gives the mean over all population responses. Grey shaded regions approximately divide the plane of eigenvalue components into regions of qualitatively different dynamics: fixed point attractor; stable spiral (bottom-right schematic); unstable spiral (top-right schematic). (C) As panel B, converted to estimates of orbital period and rate of contraction. (Note that higher imaginary eigenvalues equate to faster orbital periods, so the ordering of population responses is flipped on the x-axis compared to panel B). (D) A preparation with a visible spiral attractor in a three-dimensional projection. Each line is one of the three evoked population responses, colour-coded by time-elapsed since stimulation (grey circle). The periodicity of the evoked response is the number of loops in the elapsed time; loop magnitude corresponds to the magnitude of population activity. The approximate dominant axis of the spiral's contraction is indicated. Bottom: corresponding raster plot of one evoked response. Neurons are clustered into ensembles, and colour-coded by the change in ensemble firing rate to show the dominance of decreasing rates corresponding to the contracting loop in the projection. (E) As panel D, but for a preparation with simultaneously visible dominant spiral and minor expansion of the low-dimensional trajectory. The expansion corresponds to the small population of neurons with increasing rates.

*Figure 4 continued on next page*

*Figure 4 continued*

The following figure supplement is available for figure 4:

**Figure supplement 1.** Further properties of the spiral attractor.

neck contraction portion of the cyclical escape locomotion. We thus sought to decode the spiking of P10 directly from the low-dimensional population trajectory (*Figure 6A*).

We first confirmed that each recorded neural population did not appear to contain any motor-neurons with axons in P10, which could make the decoding potentially trivial (*Figure 6—figure supplement 1*). To then decode P10 activity, we used a statistical model that predicts the firing rate of nerve P10 at each time point, by weighting and summing the recent history (up to 100 ms) of the trajectory in the low dimensional space, and using a non-linearity to convert this weighted sum into a firing rate. We controlled for over-fitting using cross-validation forecasting: we fit the model to a 40 s window of trajectory data, and predicted the next 10 s of P10 activity (*Figure 6B*). By sliding the window over the data, we could assess the quality of the forecast over the entire recording (*Figure 6C*).

The model could accurately fit and forecast P10 activity from the low-dimensional trajectory in all nine population responses (*Figure 6D*). Emphasising the quality of the model, in *Figure 6D* we plot example forecasts of the entire P10 recording based on fitting only to the first 40 s window, each example taken from the extremes we obtained for the fit-quality metrics. Notably, in one recording the population response shutdown half-way through; yet despite the model being fit only to the 40 s window containing strong oscillations, it correctly forecasts the collapse of P10 activity, and its slight rise in firing rate thereafter. Thus, the low dimensional trajectory of the periodic attractor appears to directly encode muscle commands for movement.

To confirm this, we asked whether the encoding – as represented by the P10 activity – was truly low-dimensional. The successful decoding of future P10 activity was achieved despite needing only 3–5 embedding dimensions to account for 80% variance in the population activity for these nine recordings (*Figure 6—figure supplement 2*). Increasing the number of embedding dimensions to account for 90% variance, at least doubling the number of embedding dimensions, did not improve the forecasts of P10 activity (*Figure 6—figure supplement 2*). These results suggest that the low dimensional population trajectory is sufficient to encode the locomotion muscle commands.

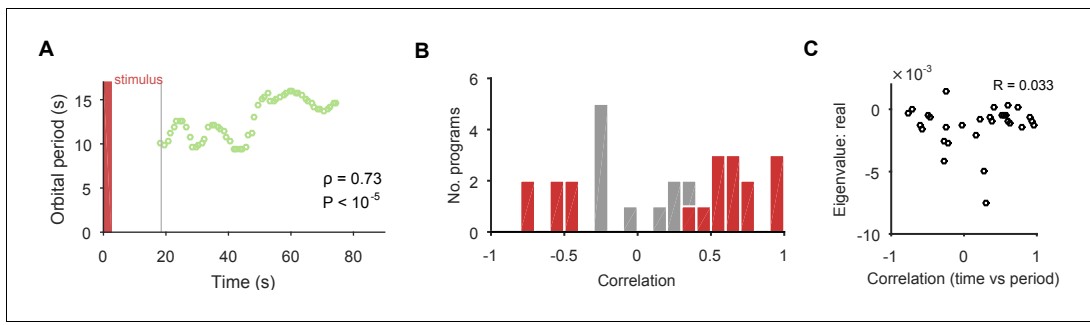

**Figure 5.** The spiral attractor dissociates changes in oscillation period and firing rate. (**A**) Example of a change in the local estimate of the periodic orbit over a population response; here, slowing over time ($n = 57$ points are each averages over a 5 s sliding window; $\rho$ is weighted Spearman's rank correlation - see Materials and methods; $P$ from a permutation test). Changes in the periodic orbit were assessed only after coalescence to the manifold (grey line). (**B**) Histogram of correlations between time elapsed and local estimate of the periodic orbit for each population response (positive: slowing; negative: speeding up). Red bars correspond to population responses with $P<0.01$ (permutation test). Number of local estimates ranged between 31 and 72 per population response. (**C**) Relationship between the change in periodic orbit over time and the rate of contraction for each population response (Pearson's R; $n = 30$ responses).

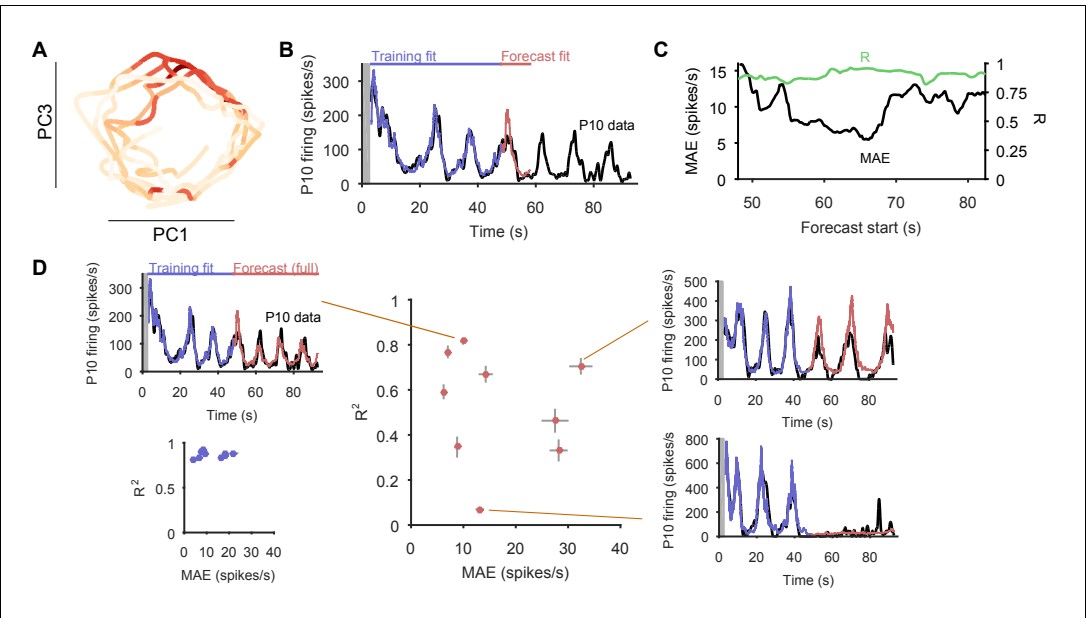

**Figure 6.** Motor output can be decoded directly from the low-dimensional trajectory of population activity. (**A**) An example two-dimensional projection of one population's response trajectory, color-coded by simultaneous P10 firing rate. In this example pair of dimensions, we can see nerve P10 firing is phase-aligned to the periodic trajectory of population activity. (**B**) Example fit and forecast by the statistical decoding model for P10 firing rate. Grey bar indicates stimulation time. (**C**) For the same example P10 data, the quality of the forecast in the 10 s after each fitted 40 s sliding window. Match between the model forecast and P10 data was quantified by the fits to both the change (R: correlation coefficient) and the scale (MAE: median absolute error) of activity over the forecast window. (**D**) Summary of model forecasts for all nine population responses with P10 activity (main panel). Dots and lines show means $\pm 2$ s.e.m. over all forecast windows ($N = 173$). Three examples from the extremes of the forecast quality are shown, each using the fitted model to the first 40 s window to forecast the entire remaining P10 time-series. The bottom right example is from a recording in which the population response apparently shutdown half-way through. Inset, lower left: summary of model fits in the training windows; conventions as per main panel.

The following figure supplements are available for figure 6:

**Figure supplement 1.** Ruling out P10 motorneurons in the recorded population.

**Figure supplement 2.** Increasing the dimensionality of state-space did not improve the P10 decoding model.

## Variable neuron participation in stable motor programs

If the low-dimensional trajectory described by the joint activity of the population just is the motor program for locomotion, then how crucial to this program are the firing of individual neurons (*Katz et al., 2004*; *Carmena et al., 2005*; *Hill et al., 2012*; *Huber et al., 2012*; *Carroll and Ramirez, 2013*; *Hill et al., 2015*)? Having quantified the motor program as the low-dimensional activity trajectory, we could uniquely ask how much each neuron participated in each evoked program. We quantified each neuron's *participation* as the absolute sum of its weights on the principal axes (eigenvectors): large total weights indicate a dominant contribution to the low-dimensional trajectory, and small weights indicate little contribution. So quantified, participation is a contextual measure, giving the contribution to the population trajectory of both a neuron's firing rate and its synchrony with other neurons, relative to the rate and synchrony of all other neurons in the population (*Figure 7—figure supplement 1*).

Every population response had a long-tailed distribution of participation (*Figure 7A*), indicating that a minority of neurons dominated the dynamics of any given response. Nonetheless, these neurons were not fixed: many with high participation in one population response showed low participation in another (*Figure 7B,C*). To rule out noise effects on the variability of participation (for

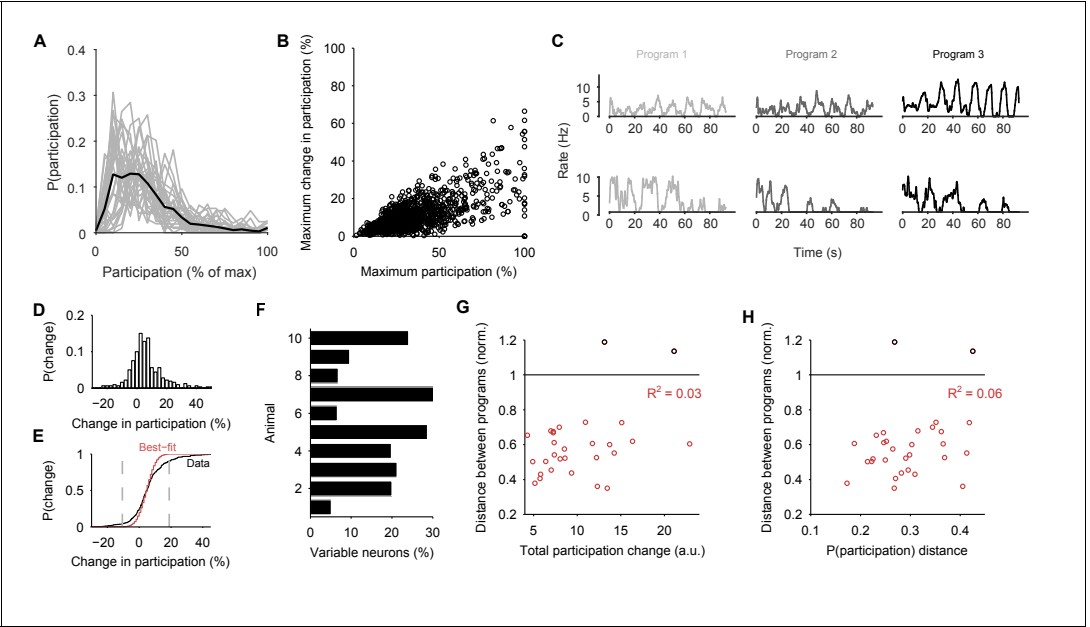

**Figure 7.** Single neuron participation varies within and between evoked locomotion bouts. (A) Distributions of single neuron participation per evoked population response. We plot the distribution of participation for all neurons in a population (grey line), expressed as a percentage of the maximum participation in that population's response. Black line gives the mean over all 30 population responses. (B) Change in participation between evoked locomotion bouts. Each dot plots one neuron's maximum participation over all three evoked population responses, against its maximum change in participation between consecutive responses ($n = 1131$ neurons). (C) Two example neurons with variable participation between responses, from two different preparations. (D) Distribution of the change in participation between responses for one preparation. (E) Detecting strongly variable neurons. Gaussian fit (red) to the distribution of change in participation (black) from panel D. Neurons beyond thresholds (grey lines) of mean $\pm$3SD of the fitted model were identified as strongly variable. (F) Proportion of identified strongly variable neurons per preparation. (G) Distance between pairs of population responses as a function of the total change in neuron participation between them. Each dot is a pair of responses from one preparation; the distance between them is given as a proportion of the mean distance between each response and a random projection (<1: closer than random projections), allowing comparison between preparations (*Figure 3C*). Black dots are excluded outliers, corresponding to the pairs containing response 1 in preparation 4 with apparent chaotic activity (*Figure 3—figure supplement 1*). (H) Distance between pairs of population responses as a function of the distance between the distributions of participation (panel A). Conventions as for panel G.

The following figure supplements are available for figure 7:

**Figure supplement 1.** Participation captures both rate and synchrony effects.

**Figure supplement 2.** Testing for an invariant central pattern generator.

example, due to the finite duration of recording), we fitted a noise model to the change in participation separately for each preparation (*Figure 7D,E*). Every preparation's pedal ganglion contained neurons whose change in participation between responses well-exceeded that predicted by the noise model (*Figure 7F*). Consequently, the contribution of single neurons was consistently and strongly variable between population responses in the same preparation.

We also tested for the possibility that hidden within the variation between programs is a small core of neurons that are strongly participating, yet invariant across programs. Such a core of phasically active neurons may, for example, form the basis of a classical central pattern generator. However, in our observed portion of the ganglion we found no evidence for a core of strongly participating, invariant, and phasically active neurons across the preparations (*Figure 7—figure supplement 2*).

These data show that a neuron's role within the locomotion motor program is not fixed, but leave open the question of whether single neuron variability causes variation in the program itself. In our analysis, variation between sequentially-evoked population responses is quantified by the distance

between their low-dimensional projections (as in *Figure 3C*). We found that the distance between a pair of population responses did not correlate with either the total change in neuron participation between the two responses (*Figure 7G*) or the distance between their participation distributions (*Figure 7H*). The execution of the motor program is thus robust to the participation of individual neurons.

## Participation maps identify potential locations of the pattern generator network

To get some insight into the physical substrate of the attractor, we plotted maps of the participation of each neuron in each preparation. We found that neurons with strong participation across the three evoked population responses were robustly located in the caudo-lateral quadrant of the ganglion (*Figure 8A,B*). Maps of the right ganglion also indicated strong participation in the rostro-medial quadrant; due to the low numbers of maps for each side, it is unclear whether this is a true asymmetry of the ganglia or simply reflects sampling variation. Neurons with highly variable participation between population responses (*Figure 8C,D*) were similarly found in the caudo-lateral quadrants of both ganglia. Strongly participating neurons were thus confined to specific regions of the pedal ganglion's network.

These data are consistent with a network-level distribution of the attractor, with a particularly strong contribution from the caudo-lateral quadrant. Encouragingly, from a different data-set we previously described this region as containing neural ensembles that generated a cyclical packet of neural activity, which moved in phase with activity from the neck-projecting P10 nerve (*Bruno et al., 2015*). Consequently, both those data and our new data support our hypothesis that the pattern generator for locomotion is predominantly located in the caudo-lateral network.

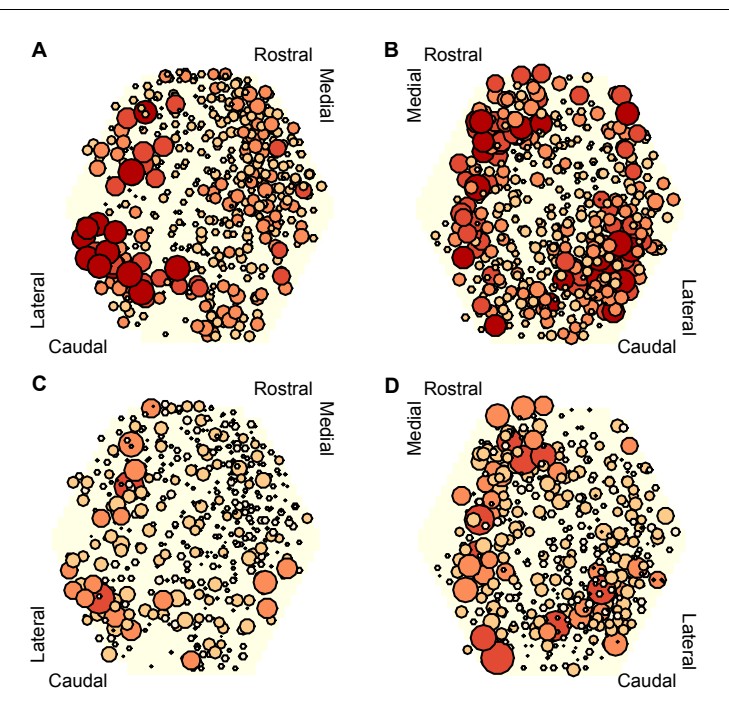

**Figure 8.** Mapping of participation in the attractor across the ganglion network. Here we plot neuron location with respect to the photodiode array (yellow hexagon). Each plot pools neurons from preparations of the left ($n = 4$ preparations) or right ($n = 4$) ganglia. **A,B** Maps of maximum participation across the three evoked population responses for left (**A**) and right (**B**) ganglion recordings. The area of each marker is proportional to the neuron's maximum participation. Neurons are colour coded (light orange to dark red) by the quintile of their participation across all preparations. **C,D** As for panels (**A,B**), but plotting the range of participation across the three evoked population responses.

## Discussion

Locomotion networks provide a tractable basis for testing theories of neural dynamics (*Lewis and Kristan, 1998*; *Briggman et al., 2005*; *Levi et al., 2005*; *Briggman and Kristan, 2006*; *Berg et al., 2007*; *Bruno et al., 2015*; *Petersen and Berg, 2016*), as they couple complex dynamics with clearly defined outputs. We took advantage of this to comprehensively test the idea that high-dimensional population activity arises from an underlying low-dimensional dynamical system: to determine what dynamical system accounts for the population activity, whether its low-dimensional signal encodes movement, and how single neuron activity relates to that signal. We showed here that *Aplysia*'s pedal ganglion contains a spiral attractor, that the low-dimensional signal it generates directly encodes muscle commands, and yet individual neurons vary in their participation in the attractor.

### A consistent low-dimensional spiral attractor

Testing the idea that high-dimensional population activity contains a low-dimensional signal has only been possible in the last decade or so, due to the necessary combination of large-scale multi-neuron recording and dimension reduction approaches (*Brown et al., 2004*; *Briggman et al., 2006*; *Cunningham and Yu, 2014*; *Kobak et al., 2016*). Landmark studies have used this combination to project high-dimensional population activity into a more tractable low-dimensional space. In this space, studies have shown how activity trajectories are different between swimming and crawling (*Briggman et al., 2005*); distinguish olfactory (*Mazor and Laurent, 2005*), auditory (*Bartho et al., 2009*), and visual (*Mante et al., 2013*) stimuli; and distinguish upcoming binary choices (*Harvey et al., 2012*). Here we have gone a step further than previous studies by not only observing such low-dimensional signals, but explicitly testing for the first time the type of dynamical system that gives rise to the low-dimensional trajectories and its consistency between animals.

Across all 30 evoked population responses examined here, there was a remarkable heterogeneity of spike-train patterns, from visually evident widespread oscillations to noisy, stuttering oscillations in a minority of neurons (*Figure 2—figure supplement 1*). Yet our analysis shows that underpinning this heterogeneity is the same dynamical system: a low-dimensional, decaying, periodic orbit. We found a remarkably consistent periodicity and rate of orbital decay across evoked responses within a preparation and between preparations. The stability of these dynamics, and the convergence of population activity to the same manifold, are all consistent with the expected behaviour of a true attractor. Our data thus suggest that only the low-dimensional system and not the high-dimensional population activity are consistent within and between nervous systems.

We advance the hypothesis that the properties of the spiral attractor fully determine the parameters of the escape gallop: its frequency, physical distance per cycle, and duration. In this hypothesis, the orbital period of the attractor determines the period of the rhythmic gallop – the sequential activity of the neurons in each orbit thus driving the sequential contraction of the muscles driving the escape gallop (*Bruno et al., 2015*). Further, the amplitude of the orbital period, corresponding to the spike rate of the neural population, could determine the strength of muscle contraction during the escape gallop, allowing control of the physical distance covered by each arching movement. Finally, the contraction rate of the attractor determines the duration of the escape: the faster the contraction rate, the shorter the escape gallop's duration. The variation of these attractor properties between animals then determines the natural variability in the escape gallop. It follows that changes to parameters of the escape gallop caused by neuromodulation should correlate with changes to the orbital period and/or contraction rate of the attractor. For example, the reported increase in gallop duration by systemic injection of serotonin (*Marinesco et al., 2004*) should correlate with a decreased contraction rate of the attractor. Future work could test this hypothesis by determining the effects of neuromodulators on the spiral attractor's properties and correlating those with readouts of the escape gallop.

Treating a neural circuit as a realisation of a dynamical system takes the emphasis away from the details of individual neurons - their neurotransmitters, their ion channel repertoire - and places it instead on their collective action. This allows us to take a Marr-ian perspective (*Marr, 1982*), which neatly separates the computational, algorithmic, and implementation levels of movement control. The computational problem here is of how to generate rhythmic locomotion for a finite duration; the algorithmic solution is a decaying periodic attractor - a spiral; and the implementation of that attractor is the particular configuration of neurons in the pedal ganglion - one of many possible

implementations (*Kleinfeld and Sompolinsky, 1988*; *Pasemann, 1995*; *Eliasmith, 2005*; *Rokni and Sompolinsky, 2012*). Indeed, a spiral attractor is potentially a general solution to the problem of how to generate a finite rhythmic behaviour.

## Insights and challenges of variable neuron participation

We saw the separation of these levels most clearly in the variable participation of the individual neurons between evoked bouts of fictive locomotion. The projection of the pedal ganglion network's joint activity into a low dimensional space captured the locomotion motor program independently of any single neuron's activity. Even the most strongly participating neurons in a given population response could more than halve their participation in other evoked responses. These results suggest that the pedal ganglion's pattern generator is not driven by neurons that are endogenous oscillators, as they would be expected to participate equally in every response. Rather, this variation supports the hypothesis that the periodic activity is an emergent property of the network.

The adaptive function of having variably participating neurons is unknown. One possibility is that, by not relying on any core set of neurons to generate rhythmic activity, the pedal ganglion's ability to generate locomotion is robust to the loss of neurons. A related possibility is that there is 'sloppiness' (*Panas et al., 2015*) in the pedal ganglion network, such that there are many possible configurations of neurons and their connections able to realise the spiral attractor that drives locomotion (*Marder et al., 2015*). Such sloppiness allows for a far more compact specification of the developmental program than needing to genetically specify the type and wiring configuration of each specific neuron.

The wide variation of single neuron participation between evoked bouts of fictive locomotion also raises new challenges for theories of neural network attractors (*Marder and Taylor, 2011*). While a variety of models present solutions for self-sustaining periodic activity in a network of neurons (*Kleinfeld and Sompolinsky, 1988*; *Eliasmith, 2005*; *Rokni and Sompolinsky, 2012*), it is unclear if they can account for the variable participation of single neurons. A further challenge is that while the variable participation of individual neurons does not affect the underlying program, clearly it takes a collective change in single neuron activity to transition between behaviours - as, for example, in the transition from galloping to crawling in *Aplysia*. What controls these transitions, and how they are realised by the population dynamics, is yet to be explored either experimentally or theoretically.

## Possible implementations of rhythmic locomotion by the pedal ganglion network

Our results nonetheless argue against a number of hypotheses for the implementation of rhythmic locomotion by the pedal ganglion. As noted above, such single neuron variability between sequential locomotion bouts argues against the generation of rhythmic activity by one or more independent neurons that are endogenous oscillators. Our results also argue against the existence of many stable periodic states in this network (*Pasemann, 1995*). Such meta-stability would manifest as changes in periodicity following perturbation. Our results show that spontaneous divergences from the attractor overwhelmingly returned to the same attractor.

How then might the pedal ganglion network implement a spiral attractor? Our data were collected from an isolated central nervous system preparation, in which the modulatory influence of neurons outside the pedal ganglion cannot be discounted (*Jing et al., 2008*). Nonetheless, as the pedal ganglion contains the central pattern generator for locomotion (*Jahan-Parwar and Fredman, 1980*), we can suggest how that generator is realised. Our results here support the hypothesis that the periodic activity is an emergent property of the ganglion's network. We know the pedal ganglion contains a mix of interneurons and motorneurons (*Fredman and Jahan-Parwar, 1980*), and that the motorneurons are not synaptically coupled (*Hening et al., 1979*), suggesting they read-out (and potentially feedback to) the dynamics of an interneuron network. An hypothesis consistent with our results here is that the ganglion contains a recurrent network of excitatory interneurons, centred on the caudo-lateral quadrant, which feed-forward to groups of motorneurons (*Bruno et al., 2015*). This recurrent network embodies the attractor, in that stimulation of the network causes a self-sustained packet of activity to sweep around it (*Bruno et al., 2015*). We see this as the periodic trajectory of joint population activity (cf *Figure 2A*, *Figure 3B*).

## Multiple periodic attractors and multi-functional circuits

Our data further suggest that the pedal ganglion network supports at least two stable states, the spontaneous activity and the stable-spiral attractor. Reaching the stable-spiral attractor from the spontaneous activity required long-duration, high-voltage pedal nerve stimulation (*Figure 1*; *Bruno et al., 2015*). In dynamical systems terms, this suggests that the spontaneous state's basin of attraction is large: most perturbations return to that state, and it takes a large perturbation to move into a different basin of attraction.

Multiple co-existing periodic attractors in a single network is also a challenge for current theories. While point attractor networks, such as Hopfield networks, can have vast number of stable states defined by different arrangements of the equilibrium activity of their neurons (*Miller, 2016*), a stable periodic attractor network typically has only two stable states: silence and periodic activity. The co-existence of stable spontaneous and periodic states in the same network suggests that something must reconfigure the network to sustain periodic activity (*Calin-Jageman et al., 2007*); otherwise, irrespective of the stimulation, the network would always return to the spontaneous state. One possibility in the pedal ganglion is that serotonin alters the effective connections between neurons: escape galloping is both dramatically extended by systemic injection of serotonin alongside tail stimulation (*Marinesco et al., 2004*), and evoked by stimulating serotonergic command neurons CC9/CC10 in the cerebral ganglion (*Jing et al., 2008*). Future experimental work should thus test the stability of the spontaneous state, and test how manipulating serotonin affects reaching and sustaining the stable-spiral attractor.

There are potentially more stable states within the pedal ganglion's network. The long-lasting crawl that follows the escape gallop is slower and omits the periodic arching of the body (*Flinn et al., 2001*). We saw three perturbations of the attractor activity that were suggestive of a transition to a different, slower periodic orbit (e.g. panel C in *Figure 3—figure supplement 2*), consistent with a transition from galloping to crawling. Such crawling is also the animal's normal mode of exploration (*Leonard and Lukowiak, 1986*), and so the 'crawling' attractor must be reachable from the spontaneous state too. *Aplysia*'s exploratory head-wave, moving its head side-to-side presumably to allow its tentacles and other head sensory organs to sample the environment (*Leonard and Lukowiak, 1986*), is also controlled by motorneurons in the pedal ganglion (*Kuenzi and Carew, 1994*). Previous studies of the *Aplysia*'s abdominal ganglion (*Wu et al., 1994*), the leech segmental ganglion (*Briggman and Kristan, 2006*), and the crustacean stomatogastric ganglion (reviewed in *Marder and Bucher, 2007*) have described multi-functional networks in which the same neurons are active in different motor behaviours. Our work here is consistent with the hypothesis that such multi-function is due to the neurons participating in different attractors realised by same network (*Briggman and Kristan, 2008*). Further work is needed to map the pedal ganglion network's dynamics to the full range of *Aplysia* motor behaviour.

## Outlook

Finding and quantifying the attractor required new analytical approaches. We introduce here the idea of using recurrence analysis to solve two problems: how to identify periodic activity in a high-dimensional space; and how to identify when the recorded system is and is not on the manifold of the attractor. By extracting the times when the population activity is on the manifold, we could then quantify and characterise the attractor, including identifying transient perturbations, and estimating changes in orbital period. Crucially, these manifold-times let us further introduce the idea of using linear models as a statistical estimator, to identify the type of attractor, and compare the detected attractor's parameters within and between preparations. Our analysis approach thus offers a roadmap for further understanding the dynamics of neural populations.

There is rich potential for understanding spontaneous, evoked or learning-induced changes in the dynamics of populations for movement control. The dynamics of movement control populations transition between states either spontaneously or driven by external input (*Briggman et al., 2005*; *Levi et al., 2005*). Our recurrence approach allows both the detection of transitions away from the current state (*Figure 3*) and the characterisation of the attractor in the new state. For learning, taking an attractor-view allows us to distinguish three distinct ways that short (*Stopfer and Carew, 1988*; *Katz et al., 1994*; *Hill et al., 2015*) or long-term (*Hawkins et al., 2006*) plasticity could change the underlying attractor: by changing the shape of the manifold; by changing the rate of

movement of the low-dimensional signal on the manifold; or by changing the read-out of the manifold by downstream targets. Such insights may contribute to the grand challenge of systems neuroscience, that of finding simplifying principles for neural systems in the face of overwhelming complexity (*Koch, 2012*; *Yuste, 2015*).

## Materials and methods

### Data and code availability
Bandpassed optical data, spike-sorted data, and available P10 nerve recordings are hosted on CRCNS.org at: 10.6080/K0SN074B

All research code is available under a MIT License from (*Humphries, 2017*): https://github.com/mdhumphries/AplysiaAttractorAnalysis. A copy is archived at https://github.com/elifesciences-publications/AplysiaAttractorAnalysis.

### Imaging
Full details of the *Aplysia californica* preparation are given in *Bruno et al. (2015)*. Briefly, the cerebral, pleural and pedal ganglia were dissected out, pinned to the bottom of a chamber, and maintained at $15 - 17°C$. Imaging of neural activity used the fast voltage sensitive absorbance dye RH-155 (Anaspec), and a 464-element photodiode array (NeuroPDA-III, RedShirtImaging) sampled at 1600 Hz. Optical data from the 464 elements were bandpass filtered in Neuroplex (5 Hz high pass and 100 Hz low pass Butterworth filters), and then spike-sorted with independent component analysis in MATLAB to yield single neuron action potential traces (the independent components), as detailed in (*Hill et al., 2010*). Rhythmic locomotion motor programs were elicited using 8V 5 ms monophasic pulses delivered at 20 Hz for 2.5 s via suction electrode to pedal nerve 9. A separate suction electrode was attached to pedal nerve 10 to continuously monitor the locomotion rhythm (*Xin et al., 1996*). Evoked activity could last for many minutes; our system allowed us to capture a maximum of $\approx 125$ s, divided between 30 s of spontaneous activity and 95 s of evoked activity. The stimulation protocol (*Figure 1B*) used short (15 min) and long (60 min) intervals between stimulations, as the original design also sought effects of sensitisation.

### Spike-train analysis
Power spectra were computed using multi-taper spectra routines from the Chronux toolbox (*Bokil et al., 2010*). We computed the power spectrum of each neuron's spike-train post-stimulation, and plot means over all spectra within a recorded population, and the mean over all mean spectra. We computed the spike-density function $f(t)$ for each neuron by convolving each spike at time $t_s$ with a Gaussian $G : f(t) = \sum_{t_0 < t_s < t_1} G(t_s) / \int_{t_0}^{t_1} G(t^*) dt^*$, evaluated over some finite window between $t_0$ and $t_1$ (see *Szucs, 1998*). We set the window to be $\pm 5\sigma$, and evaluated the convolution using a time-step of 10 ms. We defined the standard deviation $\sigma$ of the Gaussian by the median inter-spike interval of the population: $\sigma = \{\text{median ISI in population}\}/\sqrt{12}$ (see *Humphries, 2011*).

To visualise the entire population's spiking activity (*Figure 1C*), we cluster neurons by the similarity of their firing patterns using our modular deconstruction toolbox (*Bruno et al., 2015*). Different dynamical types of ensembles were identified by the properties of their autocorrelograms: tonic, oscillator, burster, or pauser - see (*Bruno et al., 2015*) for details. We also assigned each neuron in the ensemble the same dynamical label, which we use in the analysis of *Figure 7—figure supplement 2*. To demonstrate the firing rate change of each ensemble (*Figure 4*), we first counted the number of spikes emitted by that ensemble in 20 s windows, advanced in 5 s steps from the onset of stimulation. We then correlated (Pearson's $R$) the time of each window against its spike count: ensembles were classified as decreasing rate if $R < -0.2$, and increasing if $R > 0.2$.

### Model network
We used a three-neuron network to demonstrate the dynamical properties of a periodic attractor as realised by neurons (*Figure 1—figure supplement 1*). Each neuron's membrane dynamics were given by $\tau_a \dot{a}_i = -a_i(t) + c_i(t) + \sum_{j=1}^{3} w_{ji} r_j(t) - \gamma y_i(t)$, with adaptation dynamics $\tau_y \dot{y}_i = -y_i(t) + r_i(t)$, and output firing rate $r_i(t) = \max\{0, a_i(t)\}$. Weights $w_{ji} \leq 0$ give the strength of inhibitory connections

between the neurons, each of which receives a driving input $c_i$. This model, due to Matsuoka (*Matsuoka, 1985*, *1987*), generates self-sustained oscillation of network firing rates given constant scalar inputs $c_i(t) = c$, despite each neuron not being an endogenous oscillator: consequently the oscillations are an emergent property of the network. The time constants of membrane $\tau_a$ and adaptation $\tau_y$ dynamics, together with the strength of adaptation $\gamma$, determine the periodicity of the oscillations (*Matsuoka, 1985*, *1987*). Here we use $\tau_a = 0.025$ s, $\tau_y = 0.2$ s, and $\gamma = 2$; input was $c_i = 3$ throughout except where noted.

## Recurrence analysis

Low dimensional projections of the joint population activity were obtained for each program using standard principal components analysis, applied to the covariance matrix of the spike-density functions. The $d$ leading eigenvectors $W_i$ of the covariance matrix define the $d$ principal dimensions, and the $d$ corresponding eigenvalues are proportional to the variance accounted for by each dimension. The projection (the 'principal component') onto each of the chosen dimensions is given by $p_i(t) = \sum_{k=1}^{n} W_i^k f^k(t)$, where the sum is taken over all $n$ neurons in the analyzed population.

We used recurrence analysis (*Lathrop and Kostelich, 1989*; *Marwan et al., 2007*) to determine if the low-dimensional projection contained a stable periodic orbit. To do so, we checked if the low-dimensional projection $P(t) = (p_1(t), p_2(t), \ldots, p_d(t))$ at time $t$ recurred at some time $t + \delta$ in the future. Recurrence was defined as the first point $P(t + \delta) = (p_1(t + \delta), p_2(t + \delta), \ldots, p_d(t + \delta))$ that was less than some Euclidean distance $\theta$ from $P(t)$. The recurrence time of point $P(t)$ is thus $\delta$s. Contiguous regions of the projection's trajectory from $P(t)$ that remained within distance $\theta$ were excluded. Threshold $\theta$ was chosen based on the distribution of all distances between time-points, so that it was scaled to the activity levels in that particular program. Throughout we use the 10% value of that distribution as $\theta$ for robustness to noise; similar periodicity of recurrence was maintained at all tested thresholds from 2% upwards (*Figure 2—figure supplement 2*).

We checked every time-point $t$ between 5 s after stimulation until 10 s before the end of the recording (around 7770 points per program), determining whether it was or was not recurrent. We then constructed a histogram of the recurrence times using 1 s bins to detect periodic orbits (*Figure 2E*): a large peak in the histogram indicates a high frequency of the same delay between recurrent points, and thus a periodic orbit in the system. All delays less than 5 s were excluded to eliminate quasi-periodic activity due to noise in otherwise contiguous trajectories. Peaks were then defined as contiguous parts of the histogram between empty bins, and which contained more than 100 recurrent points. Programs had between one and four such periodic orbits. The peak containing the greatest number of recurrent points was considered the dominant periodic orbit of the program; the majority of programs had more than 50% of their recurrent points in this peak (blue-scale vectors in *Figure 2E*). The mean orbit period of the program was then estimated from the mean value of all recurrence times in that peak.

We measured the attractor's stability as the percentage of all points that were in periodic orbits. Evolving dynamics of each program were analysed using 5 s sliding windows, advanced in steps of 1 s. We defined the 'coalescence' time of the attractor as the mid-point of the first window in which at least 90% of the points on the trajectory were recurrent.

## Testing convergence to the same manifold

To determine if sequentially-evoked programs had the same manifold, we determined how closely the trajectories of each pair of programs overlapped in the low-dimensional space. We first projected all three programs from one preparation onto the principal axes of the first program, to define a common low-dimensional space. For each pair of programs $(A, B)$ in this projection, we then computed the Haussdorf distance between their two sets of recurrent points, as this metric is suited to handling tests of closeness between irregularly shaped sets of points. Given the Euclidean distances $\{d(A, B)\}$ from all recurrent points in $A$ to those in $B$, and vice-versa $\{d(B|A)\}$, this is the maximum minimum distance needed to travel from a point in one program to a point in the other (namely $\max\{\min\{d(A, B)\}, \min\{d(B, A)\}\}$). To understand if the resulting distances were close, we shuffled the assignment of time-series to neurons, then projected onto the same axes giving shuffled programs $A^*$, $B^*$. These give the trajectories in the low-dimensional space determined by just the firing patterns of neurons. We then computed the shuffled Haussdorf distance

$\max\{\min\{d(A, B^*)\}, \min\{d(B, A^*)\}\}$. The shuffling was repeated 100 times. Mean $\pm$ 2SEM of the shuffled distances are plotted in (*Figure 3C*); the error bars are too small to see.

To check the robustness of the convergence to the same manifold, we repeated this analysis starting from a common set of principal axes for the three programs, obtained using principal component analysis of their concatenated spike-density functions. We plot the results of this analysis in panel A of *Figure 3—figure supplement 1*.

As a further robustness control, we sought evidence of the manifold convergence independent of any low-dimensional projection. We made use of the idea that if neurons are part of sequential programs on a single manifold, then the firing of pairs of neurons should have a similar time-dependence between programs (*Yoon et al., 2013*; *Peyrache et al., 2015*). For each pair of programs $(A, B)$ from the same preparation, we computed the similarity matrix $S(A)$ between the spike-density functions of all neuron pairs in $A$, and similarly for $B$, giving $S(B)$. We then computed the correlation coefficient between $S(A)$ and $S(B)$: if $A$ and $B$ are on the same manifold, so their pairwise correlations should themselves be strongly correlated. As a control we computed a null model where each neuron has same total amount of similarity as in the data, but its pairwise similarity with each neuron is randomly distributed (*Humphries, 2011*). The expected value of pairwise correlation between neurons $i$ and $j$ under this model is then $E_{ij} = s_i s_j / T$, where $(s_i, s_j)$ are the total similarities for neurons $i$ and $j$, and $T$ is the total similarity in the data matrix. For comparison, we correlated $S(A)$ with $E$, and plot these as the control correlations in *Figure 3E*.

## Testing return to the same manifold after perturbation

We detected divergences of the trajectory away from the putative manifold, indicating spontaneous perturbations of population dynamics. We first defined potential perturbations after coalescence as a contiguous set of 5 s windows when the density of recurrent points was below 90% and fell below 50% at least once. The window with the lowest recurrence density in this divergent period was labelled the divergent point. We removed all such divergent periods whose divergent point fell within two oscillation cycles of the end of the recording, to rule out a fall in recurrence due solely to the finite time horizon of the recording. For the remaining 19 divergent periods, we then determined if the population activity returned to a recurrent state after the divergent point; that is, whether the density of recurrence returned above 90% or not. The majority (17/19) did, indicating the perturbation returned to a manifold.

For those 17 that did, we then determined if the recurrent state post-divergence was the same manifold, or a different manifold. For it to be the same manifold after the spontaneous perturbation, then the trajectory before the perturbation should recur after the maximum divergence. To check this, we took the final window before the divergent period, and counted the proportion of its recurrent delays that were beyond the end of the divergent period, so indicating that the dynamics were in the same trajectory before and after the divergence. We plot this in *Figure 3H*.

## Statistical estimation of the attractor's parameters

We introduce here a statistical approach to analysing the dynamics of low-dimensional projections of neural activity time-series obtained from experiments. We first fitted a linear model around each point on the low-dimensional trajectory to capture the local dynamics. For each point $P(t)$, we took the time-series of points before and after $P(t)$ that were contiguous in time and within $2.5 \times \theta$ as its local neighbourhood; if less than 100 points met these criteria $P(t)$ was discarded. We then fitted the dynamical model $\dot{P}^* = AP^*$ that described the local evolution of the low-dimensional projection $P^*$ by using linear regression to find the Jacobian matrix $A$; to do so, we used the selected local neighbourhood time-series as $P^*$, and their first-order difference as $\dot{P}^*$. The maximum eigenvalue $\lambda = a + ib$ of $A$ indicates the dominant local dynamics (*Strogatz, 1994*), whether contracting or expanding (sign of the real part $a$ of the eigenvalue), and whether oscillating or not (existence of the complex part of the eigenvalue that is, $b \neq 0$). The other eigenvalues, corresponding to the $d - 1$ remaining dimensions, indicate other less-dominant dynamics; usually these were consistent across all dimensions (*Figure 4—figure supplement 1*). We fitted $A$ to every point $P(t)$ after the stimulation off-set, typically giving $\approx 5000$ local estimates of dynamics from retained $P(t)$. The dominant dynamics for the whole program were estimated by averaging over the real $a$ and the complex $b$ parts of the maximum eigenvalues of the models fitted to all recurrent points in the dominant periodic orbit.

The linear model's estimate of the orbit rotation period was estimated from the complex part as $\omega = 2\pi b \Delta t$, with the sampling time-step $\Delta t = 0.01$ s here. The linear model's estimate of the contraction rate is $\exp(a/\Delta t)$, which we express as a percentage.

## Tracking changes in periodicity over a program

We tracked changes in the oscillation period by first averaging the recurrence time of all recurrent points in a 5 s sliding window. We then correlated the mean time with the time-point of the window to look for sustained changes in the mean period over time, considering only windows between coalescence and the final window with 90% recurrent points. We used a weighted version of Spearman's rank to weight the correlation in favour of time windows in which the trajectory was most clearly on the periodic orbit, namely those with a high proportion of recurrent points and low variation in recurrence time. The weighted rank correlation is: given vectors $x$ and $y$ of data rankings, and a vector of weights $w$, compute the weighted mean $m = \sum_i w_i x_i / \sum_i w_i$ and standard deviation $\sigma_{xy} = \sum_i w_i (x_i - m_x)(y_i - m_y) / \sum_i w_i$, and then the correlation $\rho = \sigma_{xy} / \sqrt{\sigma_{xx}\sigma_{yy}}$. We used the weight vector: $w_i = s_i^{-1} Q_i$, where $s_i$ is the standard deviation of recurrence times in window $i$, and $Q_i$ is the proportion of recurrent points in window $i$. P-values were obtained using a permutation test with 10000 permutations.

## Decoding motor output

We decoded P10 activity from the low-dimensional trajectory of population activity using a generalised linear model. We first ruled out that any simultaneously recorded neuron was a motorneuron with an axon in nerve P10, by checking if any neurons had a high ratio of locking between their emitted spikes and spikes occurring at short latency in the P10 recording. *Figure 6—figure supplement 1* shows that no neuron had a consistent, high ratio locking of its spikes with the P10 activity.

We convolved the spikes of the P10 recording with a Gaussian of the same width as the spike-density functions of the simultaneously recorded program, to estimate its continuous firing rate $f_{10}$. We fitted the model $f_{10}(t) = \exp\left(\beta_0 + \sum_{i=1}^{d} \sum_{h=1}^{m} \beta_{i,h} P_i(t-h)\right)$ to determine the P10 firing rate as a function of the past history of the population activity trajectory. Using a generalised linear model here allows us to transform the arbitrary co-ordinates of the $d$-dimensional projection $P(t)$ into a strictly positive firing rate. Fitting used glmfit in MATLAB R2014. To cross-validate the model, we found the coefficients $\beta$ using a 40 s window of data, then forecast the P10 firing rate $f_{10}^*$ using the next 10 s of population recording data as input to the model. Forecast error was measured as both the median absolute error and the correlation coefficient $R$ between the actual and forecast P10 activity in the 10 s window. The fitting and forecasting were repeated using a 1 s step of the windows, until the final 40 s + 10 s pair of windows available in the recording.

We tested activity histories between 50 and 200 ms duration, with time-steps of 10 ms, so that the largest model for a given program had $d \times 20$ coefficients. These short windows were chosen to rule out the contributions of other potential motorneurons in the population recording that would be phase offset from neck contraction (as 200 ms is 2% of the typical period). All results were robust to the choice of history duration, so we plot results using history durations that had the smallest median absolute error in forecasting for that program.

## Single neuron participation

We quantified each neuron's participation in the low-dimensional projection as the L1-norm: the absolute sum of its weights on the principal axes (eigenvectors) for program $m: \rho_i^m = \sum_{j=1}^{d} \left| \lambda_j^m W_j^m(i) \right|$, where the sum is over the $d$ principal axes, $W_j^m(i)$ is the neuron's weight on the $j$th axis, and $\lambda_j^m$ is the axis' corresponding eigenvalue. Within a program, participation for each neuron was normalised to the maximum participation in that program. To fit a noise model for the variability in participation between programs, we first computed the change in participation for each neuron between all pairs of programs in the same preparation. We then fit a Gaussian model for the noise, using an iterative maximum likelihood approach to identify the likely outliers; here the outliers are the participation changes that are inconsistent with stochastic noise. In this approach, we compute the mean and variance of the Gaussian from the data, eliminate the data-point furthest from the estimate of the mean, re-estimate the mean and variance, and compute the new log likelihood of the Gaussian model

without that data-point. We iterate elimination, re-estimation, and likelihood computation until the likelihood decreases. The final model (mean and variance) found before the decrease is then the best-fit Gaussian model to the bulk of the data. Neurons whose maximum change in participation exceeded a threshold of the mean ±3SD of that best-fit model were then considered 'strongly variable' neurons.

We asked whether the variation in low-dimensional dynamics of sequentially-evoked programs was a consequence of the degree of variation in single neuron participation. Between a pair of consecutively evoked programs, we quantified the variation in their low dimensional dynamics as the Hausdorff distance between them, normalised by the mean distance between their random projections. This normalisation allowed us to put all programs on a single scale measuring the closeness relative to random projections, such that one indicates equivalence to a random projection, <1 indicates closer than random projections, and >1 indicates further apart then random projections. For a given pair of programs, we quantified the variability of individual neurons' participation in two ways: by summing the change in participation of each neuron between the programs; and by computing the Hellinger distance between the two distributions of participation (one distribution per program).

## Participation maps

Each neuron's (x,y) location on the plane of the photodiode array could be estimated from the weight matrix from the independent component analysis of the original 464 photodiode time-series; see (*Bruno et al., 2015*) for full details. We were able to reconstruct locations for all neurons in 8 of the 10 recorded preparations; for the other two preparations, partial corruption of the original spike-sorting analysis data prevented reconstructions of some neuron locations in one; for the other, we could not determine on what side it was recorded. We merged all left or right ganglion recordings on to a common template of the photodiode array. The marker sizes and colour codes for each neuron were proportional to the normalised maximum participation of that neuron (*Figure 8A,C*) and to the range of normalised maximum participation across the three programs (*Figure 8B,D*).

## Acknowledgements

We thank R Petersen and B Mensh for comments on drafts, A Singh for suggesting the Hausdorff distance, and J Wang for technical assistance. MDH was supported by a Medical Research Council Senior non-Clinical Fellowship. WF was supported by NIH R01NS060921 and NSF 1257923. AB was supported by NIH F31NS079036.

## Additional information

### Funding

| Funder | Grant reference number | Author |
|---|---|---|
| Medical Research Council | MR/J008648/1 | Mark D Humphries |
| National Institutes of Health | R01NS060921 | William N Frost |
| National Science Foundation | 1257923 | William N Frost |
| National Institutes of Health | F31NS079036 | Angela M Bruno |

The funders had no role in study design, data collection and interpretation, or the decision to submit the work for publication.

### Author contributions

AMB, Conceptualization, Data curation, Investigation, Methodology, Writing—review and editing; WNF, Conceptualization, Supervision, Funding acquisition, Writing—review and editing; MDH, Conceptualization, Software, Formal analysis, Visualization, Methodology, Writing—original draft

### Author ORCIDs

Mark D Humphries, http://orcid.org/0000-0002-1906-2581

## Additional files

### Major datasets

The following dataset was generated:

| Author(s) | Year | Dataset title | Dataset URL | Database, license, and accessibility information |
|---|---|---|---|---|
| Angela M Bruno, William N Frost | 2017 | apl-1: Voltage sensitive dye recording of multiple bouts of escape locomotion in the pedal ganglion of Aplysia californica | http://dx.doi.org/10.6080/K0SN074B | Publicly available from Collaborative Research in Computational Neuroscience (accession no. K0SN074B) |

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
