## [Decision Letter]

Thank you for submitting your article "A spiral attractor network drives rhythmic locomotion" for consideration by *eLife*. Your article has been reviewed by three peer reviewers, one of whom, Jan-Marino Ramirez (Reviewer #1), is a member of our Board of Reviewing Editors and the evaluation has been overseen by Eve Marder as the Senior Editor. The following individuals involved in review of your submission have agreed to reveal their identity: Rune W Berg (Reviewer #2).

The reviewers have discussed the reviews with one another and the Reviewing Editor has drafted this decision to help you prepare a revised submission.

Summary:

The authors explore the population dynamics that controls fictive locomotion in the pedal ganglion network of *Aplysia*. The locomotor activity is a relatively simple escape response, which is elicited by a brief electrical stimulation to the tail nerve. The use of fast voltage sensitive dyes and optical imaging allows the authors to gain insights into the activity of individual neurons, while simultaneously recording the activity from 120-180 neurons, which represents approximately 10% of the entire ganglion population. This unique preparation is ideal to address fundamental questions in neuronal networks. In particular the question of how populations of neurons control behavior is difficult to address in larger networks, such as those of the mammalian nervous system.

The authors find that the population dynamics can best be described as an attractor dynamics (spiral attractor) with a low dimension, whereas the higher dimensional activity is variable from trial to trial as well as from animal to animal and therefore not attributed the same importance in the generation of the motor pattern. Interestingly, the authors found that the individual participation in the dynamics could change substantially from bout to bout, while the population dynamics was more stable. This is consistent with the property of mammalian networks in which single neurons show also large variability, while the output is rather regular (see e.g. Carroll and Ramirez, 2013).

There are several other important and interesting findings in this study. This small network investigation is simple enough to restrict the unknown variables yet rich enough to address interesting questions of neuronal network behind motor behavior. It is truly a rare and noteworthy study and the authors go through great efforts to analyze and interpret the complexity that even such a simple animal has. The paper is well written, but it would help to try to further simplify the language and better explain the terms used. The study is innovative as the authors provide tools for analysis, in particular the recurrence analysis. Thus, this study will undoubtedly inspire future investigation in attractor dynamics of neuronal networks.

Essential revisions:

1) The study is well written, but it needs to be adjusted to a more general readership. The method is innovative, but it needs to be better explained, in particular for a reader who is not familiar with the terminology used in this study. The authors should provide e.g. a better explanation in the text how eigenvalues for the population recordings were calculated.

2) We would be interested to hear some of your answers to the questions raised below. The authors could consider addressing these questions in the Discussion section, since the general readership might arrive at similar questions. But, we won't insist that the authors include each of the following issues in the text.

2a) Please comment: There seems to be only one attractor in the state space – or two if you count the spontaneous state, which seems rather trivial, as an attractor. Although this arrangement would provide robustness to the system and prevent unintended pathological activities, it seems to be contrary to the general notion of neuronal networks having multiple stable states – e.g. the classical Hopfield memory networks. Further, it was documented in an early report that (even) the neuronal network in the *Aplysia* abdominal ganglion could participate in at least 3 different behaviors, see

J-Y Wu, LB Cohen, CX Falk, "Neuronal activity during different behaviors in *Aplysia*: a distributed organization?" Science 263, 820-823, 1994

Is the pedal ganglion network not able to perform more motor behaviors or is it not possible to induce other behaviors experimentally? I know the authors make great effort to map the basin of attraction, but can you comment more on this issue of lack of multi-stability?

2b) Please comment: What is the role of the rest of the population in bringing back the dynamics to the attractor? I don't fully see to what extent the perturbation affects all the neurons, or only some. Could it be that the response to the perturbation (Figure 1) and the decay back to the attractor is really the alignment of the recorded/perturbed neurons with the rest of the unperturbed population?

2c) Please comment: Discussion and other places: "Testing the idea that high-dimensional population activity contains a low-dimensional signal […]" How would the activity look if it were a high dimensional activity without a low dimensional signal, i.e. the contrary situation? Would it just be random chaotic spiking? Would it even be possible to have a rhythmic low dimensional motor output without having that same low-dimensional component in the neuronal population?

2d) Emergence. The authors discuss in a couple of places the interesting and profound issue:

"These results suggest that the pedal ganglion's pattern generator is not driven by neurons that are endogenous oscillators, as they would be expected to participate equally in every response. Rather, this variation supports the hypothesis that the periodic activity is an emergent property of the network. "

However, in their model the network oscillation frequency is set by the membrane time constant of the single neuron. This seems at odds with the notion that the rhythm is an emergent network phenomenon. Usually "emergence" is a property that cannot be identified on the single cell level. And if so, why is the network oscillation so robust and independent on the participation of individual neuron? Is the network oscillation frequency some sort of weighted average of the time constants of different cells? This is a very important issue and has caused considerable discussion e.g. in the field of mammalian respiration, and thus, it would be great to include your considerations in the discussion. We will further elaborate on your conclusion with regards to the emergent property below.

3) The authors use recurrence analysis, which is adapted from the analysis of dynamical systems, and they provide some evidence that population activity in the pedal ganglion fulfills some criteria of attractor dynamics: A) the cyclical periodic orbit is rapidly reached from a more diverse state during the initial stimulation. B) After spontaneous perturbations, the system recovers rapidly returning to the manifold. Moreover, C) they show that across the entire recorded neuronal population, individual neurons can change in the weight by which they contribute to the low dimensional state. Based on C), they conclude that the low dimensional dynamical system is an emergent property of the neuronal population and more robust / reproducible than the high dimensional neuronal population activity; in conclusion, a stable motor output is produced besides the variability seen in individual neurons.

The use of recurrence analysis for single cell resolution neuronal population data is innovative. While this manuscript will be interesting to a large community of neuroscientists, the manuscript needs some clarifications. Most importantly, C) is a very strong statement that really would change views of how population dynamics in the pedal ganglion work; however, C) is not sufficiently supported by the analysis provided and thus could be misleading. Additional analyses should be done to either revise or solidify this conclusion.

4) Along these lines: there might be an oscillatory sub-population of neurons, or even smaller CPG unit, that generates in a very robust and reproducible manner a recurrent and periodic pattern; this view is already somewhat supported in Figure 8. In addition, other pedal ganglion neurons contribute to a more complex overall population state by exhibiting more variable activity patterns, that might be transiently recruited to this recurrently active subpopulation. If this is the case, the main statement (C) does not hold and should be revised.

5) If the dataset in Figure 1 is representative, it is obvious that PCA will lead to at least two modes that describe a dominant periodic orbit. The authors should provide more direct quantification of what type of neurons have the most variable contributions and in which way. In their previous work, the authors devised a method to classify neuronal subpopulations as non-oscillators, oscillators, bursters, pausers (Bruno et al., 2015); see also Figure 1. First, the authors should perform their analyses on these subpopulations separately; one might expect that the oscillatory neurons show much less variable contributions and that the variability comes from the other neuronal classes. Or is there even a very small set of highly non-variable neurons? In this case, one could argue that recurrence, periodicity and stable motor output generation is not an emergent property of the pedal ganglion but the robust output of a smaller subpopulation / CPG subunit.

6) PCA results, especially in the higher dimensions can be quite variable from dataset to dataset, or even within datasets, when performed on subsections. The authors use the eigenvectors to measure each neuron's contribution to the low dimensional signal. However, as they also state, this variability can come from variable signal amplitudes or variable synchrony. If signal amplitude is the main source of variability, I would argue that the main periodic (or recurrent) feature of these neurons is robust and preserved throughout the 3 programs in each recording; it just appears variable in the eigenvectors. So, the authors should show that the main source of variability is indeed variable synchrony.

7) An essential aspect of recurrence analysis described in Marwan et al., 2007 is the addition of time delay dimension to the data; here recurrent points are similar not only in their instantaneous distances but also in their time history. This step is not mentioned in the manuscript. Please provide an explanation why this was not needed for the analysis of their data.

8) Detailed information is lacking about how many PC dimensions were used in each analysis and dataset. Were always the top 4 dimensions used as described for Figure 2, or always as many dimensions that explain 80% of variance? Was this the same for all analyses throughout the paper? This choice makes a difference in how to interpret the data. Please provide this detailed information.

9) Please discuss what the function of variability might be. For example, is the size principle described for the Aplyisa motor system, i.e. do variable recruitment patterns are functional for different loads on the motor apparatus? Etc.

---

## [Author Response]

*Essential revisions:*

*1) The study is well written, but it needs to be adjusted to a more general readership. The method is innovative, but it needs to be better explained, in particular for a reader who is not familiar with the terminology used in this study. The authors should provide e.g. a better explanation in the text how eigenvalues for the population recordings were calculated.*

The accessibility of the reported work was always a concern for us, and prompted here by the referees we have made a number of changes to the text:

I) We revised the opening paragraph of the Introduction to motivate better the approach of looking at a low dimensional system for the general reader..

II) We have extended Figure 1 to include a further schematic and fuller definitions of the terminology used. In particular, we now illustrate and define "trajectory"; and illustrate and define "manifold".

III) We have revised the opening paragraph in subsection “Joint population activity meets the conditions for a periodic attractor”, this now more clearly defines the expectations of a manifold for an attractor, and better motivates the analysis then done.

IV) We have added a paragraph of explanation to subsection “Heterogenous population activity arises from a common attractor” on how the local linear model was fit and how the eigenvalues were derived from it.

V) We have extended the Discussion (see responses below), and divided it into titled subsections, to better orient the reader.

*2) We would be interested to hear some of your answers to the questions raised below. The authors could consider addressing these questions in the Discussion section, since the general readership might arrive at similar questions. But, we won't insist that the authors include each of the following issues in the text.*

*2a) Please comment: There seems to be only one attractor in the state space – or two if you count the spontaneous state, which seems rather trivial, as an attractor. Although this arrangement would provide robustness to the system and prevent unintended pathological activities, it seems to be contrary to the general notion of neuronal networks having multiple stable states – e.g. the classical Hopfield memory networks. Further, it was documented in an early report that (even) the neuronal network in the Aplysia abdominal ganglion could participate in at least 3 different behaviors, see*

*J-Y Wu, LB Cohen, CX Falk, "Neuronal activity during different behaviors in Aplysia: a distributed organization?" Science 263, 820-823, 1994*

*Is the pedal ganglion network not able to perform more motor behaviors or is it not possible to induce other behaviors experimentally? I know the authors make great effort to map the basin of attraction, but can you comment more on this issue of lack of multi-stability?*

These are all excellent questions.

A periodic attractor network and a Hopfield-style point attractor network are very different beasts. A point attractor network's stable states are defined by the number of configurations of stable, unchanging activity amongst its neurons. In the simple networks of Hopfield, Amit, Grossberg and others, these configurations are exponential in the number of neurons, as different inputs recall a different configuration of neuron activity. By contrast, a periodic attractor network has a stable states defined by repeatedly changing activity amongst its neurons. An enumeration of its stable states is defined by the number of periodic trajectories that can arise given different inputs. This can be as low as 1, with the same periodic trajectory arising for any given input.

Separately, a periodic attractor network can of course support more than one stable periodic trajectory if some element of that network is changed (e.g. the strength of connections between neurons): thus multiple attractors can exist in the same physical network if the properties of that network can be modulated. Just as we suspect happens with serotonin in the pedal ganglion.

The pedal ganglion does indeed support more than one behavior. As we touched on in the manuscript, minimally it supports both the escape gallop and the normal crawl, which have different sequences of muscle contractions. We now mention that it may also drive the head-wave of *Aplysia*.

We have brought all the above points together in a new Discussion section titled "Multiple periodic attractors and multifunctional circuits", including appropriate new references.

*2b) Please comment: What is the role of the rest of the population in bringing back the dynamics to the attractor? I don't fully see to what extent the perturbation affects all the neurons, or only some. Could it be that the response to the perturbation (Figure 1) and the decay back to the attractor is really the alignment of the recorded/perturbed neurons with the rest of the unperturbed population?*

The spontaneous perturbations we observed were the almost complete cessation of recurrence in a trajectory that accounted for at least 80% of the co-variation of the (observed) population activity. Consequently, the perturbations were population-wide, as illustrated in the three raster plots in Figure 3—figure supplement 2. The return to the attractor is not necessarily caused by any other group of neurons, but is simply the response of the system to the cessation of a transient change to its input (Figure 1—figure supplement 1).

We have now commented on this in subsection “Joint population activity meets the conditions for a periodic attractor”.

*2c) Please comment: Discussion and other places: "Testing the idea that high-dimensional population activity contains a low-dimensional signal…" How would the activity look if it were a high dimensional activity without a low dimensional signal, i.e. the contrary situation? Would it just be random chaotic spiking? Would it even be possible to have a rhythmic low dimensional motor output without having that same low-dimensional component in the neuronal population?*

High-dimensional activity without a corresponding low-dimensional signal can only arise with random, uncorrelated firing of each neuron. By contrast, chaotic spiking is not random: there is co-variation between neurons, but without any sustained periodicity (as in our example of putative chaotic spiking in Figure 3—figure supplement 1). A chaotic system will thus also have a low-dimensional signal (it’s just that the low-dimensional signal has fractional dimensions for a chaotic system).

Any co-variation between neurons reduces the dimensionality of the space needed to describe the population. So the key questions are i) how low a number of dimensions is needed, to understand the redundancy in the population activity (here we see a factor of 10 reduction in the number of dimensions) and ii) how low is relevant to the behavior? We provide evidence that the very low number of dimensions is relevant through the use of the GLM decoder to reconstruct P10 activity from the low-dimensional trajectory.

*2d) Emergence. The authors discuss in a couple of places the interesting and profound issue:*

*"These results suggest that the pedal ganglion's pattern generator is not driven by neurons that are endogenous oscillators, as they would be expected to participate equally in every response. Rather, this variation supports the hypothesis that the periodic activity is an emergent property of the network. "*

*However, in their model the network oscillation frequency is set by the membrane time constant of the single neuron. This seems at odds with the notion that the rhythm is an emergent network phenomenon. Usually "emergence" is a property that cannot be identified on the single cell level. And if so, why is the network oscillation so robust and independent on the participation of individual neuron? Is the network oscillation frequency some sort of weighted average of the time constants of different cells? This is a very important issue and has caused considerable discussion e.g. in the field of mammalian respiration, and thus, it would be great to include your considerations in the discussion. We will further elaborate on your conclusion with regards to the emergent property below.*

The fact that the time constants constrain the oscillation is not at odds with an emergent system. An emergent property of a system is one that the collection of units has that the individual units do not. But those properties are of course physically constrained by the physical properties of the units. The canonical example is flocking birds. The flock flies differently to an individual bird, with sudden changes of direction, massing and spreading. But the flock's behavior is constrained by physical properties of the bird species – its airspeed (limiting the flock’s velocity) and its wingspan (limiting its compression).

In our toy model attractor network, the single units are not oscillators. Given constant input, they transiently respond then adapt to a steady-state. The oscillation is a property of the network, not the units. But of course the possible frequencies of oscillation are constrained by time constant of and the delays between the neurons – for example, the longer the transmission delay between the connected neurons, so the lower is the upper limit of possible oscillation frequencies. Exactly what frequency emerges from the network thus depends heavily on the details of the network, with no universal answer. In a toy model like ours with just two homogenous time constant, then the oscillation frequency is determined by them in a straightforward way (largely by the time constant of adaptation). But even as soon as we move to model with heterogenous time constants and delays, the solution to the periodicity of the attractor becomes complex (see e.g. Nevado-Holgado, Terry & Bogacz (2011) J Neurosci for solutions to periodic states in the very simple STN-GPe feedback loop within the basal ganglia).

Our model is illustrative of the main concepts in a periodic attractor network – as we note in the Discussion, current models cannot account for variation of neurons between executions of the attractor. We thus don’t know why networks are robust to variation.

Figure 1—figure supplement 1 now shows what happens with uncoupled neurons in our toy model. This illustrates emergence: without the network connections, there is no oscillation.

*3) The authors use recurrence analysis, which is adapted from the analysis of dynamical systems, and they provide some evidence that population activity in the pedal ganglion fulfills some criteria of attractor dynamics: A) the cyclical periodic orbit is rapidly reached from a more diverse state during the initial stimulation. B) After spontaneous perturbations, the system recovers rapidly returning to the manifold. Moreover, C) they show that across the entire recorded neuronal population, individual neurons can change in the weight by which they contribute to the low dimensional state. Based on C), they conclude that the low dimensional dynamical system is an emergent property of the neuronal population and more robust / reproducible than the high dimensional neuronal population activity; in conclusion, a stable motor output is produced besides the variability seen in individual neurons.*

*The use of recurrence analysis for single cell resolution neuronal population data is innovative. While this manuscript will be interesting to a large community of neuroscientists, the manuscript needs some clarifications. Most importantly, C) is a very strong statement that really would change views of how population dynamics in the pedal ganglion work; however, C) is not sufficiently supported by the analysis provided and thus could be misleading. Additional analyses should be done to either revise or solidify this conclusion.*

*4) Along these lines: there might be an oscillatory sub-population of neurons, or even smaller CPG unit, that generates in a very robust and reproducible manner a recurrent and periodic pattern; this view is already somewhat supported in Figure 8. In addition, other pedal ganglion neurons contribute to a more complex overall population state by exhibiting more variable activity patterns, that might be transiently recruited to this recurrently active subpopulation. If this is the case, the main statement (C) does not hold and should be revised.*

*5) If the dataset in Figure 1 is representative, it is obvious that PCA will lead to at least two modes that describe a dominant periodic orbit. The authors should provide more direct quantification of what type of neurons have the most variable contributions and in which way. In their previous work, the authors devised a method to classify neuronal subpopulations as non-oscillators, oscillators, bursters, pausers (Bruno et al., 2015); see also Figure 1. First, the authors should perform their analyses on these subpopulations separately; one might expect that the oscillatory neurons show much less variable contributions and that the variability comes from the other neuronal classes. Or is there even a very small set of highly non-variable neurons? In this case, one could argue that recurrence, periodicity and stable motor output generation is not an emergent property of the pedal ganglion but the robust output of a smaller subpopulation / CPG subunit.*

We have dealt with points 3) to 5) together, as they ultimately are all focused on the core question of evidence for emergence in the population activity. The referees have rightly asked whether there is evidence for a core group of stable, non-variable neurons in the population.

To an extent, we already had evidence against this idea. Such a core CPG population would be neurons that are phasically active, strongly participating in all population responses, and low in variation between population responses. Figure 7 already showed that there were very few such neurons, which would be expected to fall in the bottom half of the bottom right quadrant of that scatter plot, but where instead there is predominantly whitespace.

To further address this question, we have done the following:

1) We clustered every one of the 30 population responses into their constituent ensembles (as per our modular deconstruction approach in Bruno et al., 2015).

2) Each ensemble was then classified as oscillator, burster, tonic, or pauser according to the same auto-correlation approach as in Bruno et al., 2015.

3) Each neuron was then labelled with the class of its parent ensemble.

4) We determined how many neurons were consistently labelled across all three responses in the same preparation. We found that only a subset of the oscillator class was consistently labelled across all three responses in all ten preparations. This would be consistent with a small core of invariant phasically active neurons.

5) However, this consistent subset of oscillator neurons also showed the same wide variation in participation as the whole population. We could not identify even one neuron in some populations that was strongly participating and yet invariant.

These results are summarised in a new supplemental figure: Figure 7—figure supplement 2.

We added a paragraph to the main text (subsection “Variable neuron participation in stable motor programs”) reporting these results

*6) PCA results, especially in the higher dimensions can be quite variable from dataset to dataset, or even within datasets, when performed on subsections. The authors use the eigenvectors to measure each neuron's contribution to the low dimensional signal. However, as they also state, this variability can come from variable signal amplitudes or variable synchrony. If signal amplitude is the main source of variability, I would argue that the main periodic (or recurrent) feature of these neurons is robust and preserved throughout the 3 programs in each recording; it just appears variable in the eigenvectors. So, the authors should show that the main source of variability is indeed variable synchrony.*

Figure 7—figure supplement 1 is now a more thorough quantification of participation. It shows that independently measured changes in rate and change in synchrony between responses can correlate with the change in participation, but need not do so. These results emphasise that participation is a contextual thing: the contributions to the eigenvectors of each neuron are relative to all other neurons' contributions. Moreover, participation quantifies change across multiple dimensions of co-variation in the population's activity. So correlating it with absolute and independent measures of rate or synchrony changes is only a crude guide to what is measured. Thus we used our participation index as a summary measure of the relative contribution to the population of each neuron.

*7) An essential aspect of recurrence analysis described in Marwan et al., 2007 is the addition of time delay dimension to the data; here recurrent points are similar not only in their instantaneous distances but also in their time history. This step is not mentioned in the manuscript. Please provide an explanation why this was not needed for the analysis of their data.*

We believe the referees comments here are in reference to Takens’ embedding theorem (Eq 9 in Marwan et al., 2007). This is a remarkable result for one-dimensional time-series (such as the output of a single neuron), which allows us to treat a 1D time-series as though it were a set of >1 time-series. Namely, by constructed a high-dimensional space by adding as many dimensions as time-delay steps (t,t+1,…,t+tau), and then determining the dynamical system in that time-embedding space. However, here we already have >>1 time-series, and even after PCA we still have P > 1 dimensions: so we simply check for recurrence in the natural dimensions of the system at hand (Eq 7 in Marwan et al., 2007), rather than constructing a time-embedding set of dimensions.

*8) Detailed information is lacking about how many PC dimensions were used in each analysis and dataset. Were always the top 4 dimensions used as described for Figure 2, or always as many dimensions that explain 80% of variance? Was this the same for all analyses throughout the paper? This choice makes a difference in how to interpret the data. Please provide this detailed information.*

The referees are quite right that we omitted a single clear statement from the main text, and only implied that we used the number of dimensions needed to account for 80% variance throughout.

We now explicitly state: "Thus, throughout our analysis, we projected each evoked program into the number of embedding dimensions needed to capture at least 80% of the variance in population activity" (subsection “Joint population activity forms a low-dimensional periodic orbit”).

*9) Please discuss what the function of variability might be. For example, is the size principle described for the Aplyisa motor system, i.e. do variable recruitment patterns are functional for different loads on the motor apparatus? Etc.*

We have collated our existing points on the role of variability into a Discussion section, subsection "Insights and challenges of variable neuron participation", and extended our discussion of its potential functional roles. In particular, we now speculate that one potential role of variable participation is to build "sloppiness" into the system, such that there are many possible configurations of the network able to generate the spiral attractor.